

# A climatological view of the vertical stratification of RH, O₃ and CO within the PBL and at the interface with free troposphere as seen by IAGOS aircraft and ozonesondes at northern mid-latitudes over 1994-2016

Hervé Petetin[1], Bastien Sauvage[1], Herman G. J. Smit[2], François Gheusi[1], Fabienne Lohou[1], Romain Blot[1], Hannah Clark[3], Gilles Athier[1], Damien Boulanger[4], Jean-Marc Cousin[1], Philippe Nedelec[1], Patrick Neis[2], Susanne, Rohs[2], Valérie Thouret[1]

[1]Laboratoire d'Aérologie, Université de Toulouse, CNRS, UPS, Toulouse, France
[2]Forschungszentrum Jülich GmbH, Institut für Energie- und Klimaforschung, IEK-8 Troposphere, 52425 Jülich,
Germany
[3]IAGOS-AISBL, Brussels, Belgium
[4]Observatoire Midi-Pyrénées, Université de Toulouse, CNRS, UPS, Toulouse, France

*Correspondence to:* H. Petetin (hervepetetin@gmail.com)

**Abstract.** This paper investigates in an innovative way the climatological vertical stratification of relative humidity
(RH) and ozone (O₃) and carbon monoxide (CO) mixing ratios within the planetary boundary layer (PBL) and at the interface with the free troposphere (FT). The climatology includes all vertical profiles available at northern mid-latitudes over the period 1994-2016 in both IAGOS (In-service Aircraft for a Global Observing System) and WOUDC (World Ozone and Ultraviolet Radiation Data Centre) databases, which represents more than 90,000 vertical profiles. For all individual profiles, apart from the specific case of surface-based temperature inversions (SBIs), the PBL height
is estimated following the elevated temperature inversion (EI) method. Several features of both SBIs and EIs are analysed, including their diurnal and seasonal variations. Based on these PBL height estimates (denoted $h$), the original approach introduced in this paper consists in building a so-called PBL-referenced vertical distribution of O₃, CO and RH by averaging all individual profiles beforehand expressed as a function of $z/h$ rather than $z$ (with $z$ the altitude). Using this vertical coordinate system allows to highlight the features existing at the PBL-FT interface that would have
been smoothed otherwise.

Results demonstrate that the frequently assumed well-mixed PBL remains an exception for both chemical species. Within the PBL, CO profiles are characterized by a mean vertical stratification (here defined as the standard deviation of the CO profile between the surface and the PBL top, normalized by the mean) of 11%, with moderate seasonal and diurnal variations. A higher vertical stratification is observed for O₃ mixing ratios (18%), with stronger seasonal and
diurnal variability (from ~10% in spring/summer midday/afternoon to ~25% in winter/fall night). This vertical stratification is distributed heterogeneously in the PBL with stronger vertical gradients observed at both the surface (due to dry deposition and titration by NO for O₃; and due to surface emissions for CO) and the PBL-FT interface. These gradients vary with the season from lowest values in summer to highest ones in winter. Contrary to CO, the O₃ vertical stratification was found to vary with the surface potential temperature following an interesting bell shape with weakest
stratification for both lowest (typically negative) and highest temperatures, which could be due to a much lower O₃ dry deposition under the presence of snow.

Therefore, results demonstrate that EIs act as a geophysical interface separating air masses of distinct chemical composition and/or chemical regime. This is further supported by the analysis of the correlation of O₃ and CO mixing ratios between the different altitude levels in the PBL and FT (the so-called vertical autocorrelation). Results indeed
highlight lower correlations apart from the PBL-FT interface and higher correlations within each of the two atmospheric compartments (PBL and FT).





## 1   Introduction

As the region of the atmosphere where exchanges of momentum, water and trace chemical species occur with the Earth's surface, the planetary boundary layer (PBL) is of fundamental importance for atmospheric studies. The pollutant

concentrations at the Earth's surface are intimately linked to their vertical distribution in the entire PBL, which in turn results from a complex interaction between emissions and deposition at the surface, local chemistry, horizontal advection by the wind, vertical turbulent mixing in the PBL and exchanges with the free troposphere (FT). The vertical extent and structure of the PBL is closely linked to the turbulence. Within the PBL, the turbulence can be generated by the static instability produced by surface heating (that induces thermals of warm air rising or the advection of cold air

masses above warm surfaces) or mechanically through the wind shear at the surface or in the vicinity of jets (Stull, 1988). During the development of the daytime convective PBL, air from the FT or the residual layer (RL) is entrained into the PBL which modifies the budget of the chemical species (the entrainment flux acting as a source or a sink depending on the species, the location and the time of day). The numerous processes at stake in the PBL lead to a highly variable and complex vertical distribution of pollution.

Due to the lack of in-situ data at altitude, this vertical distribution remains only partially described and is therefore still difficult to reproduce in state-of-the-art chemistry-transport models (CTMs). This was notably demonstrated by the last Air Quality Model Evaluation International Initiative (AQMEII) and Monitoring Atmospheric Composition and Climate (MACC) model inter-comparisons on ozone ($O_3$) and carbon monoxide (CO) (e.g. Elguindi et al., 2010; Solazzo et al., 2013). More recently, Travis et al. (2017) highlighted the difficulty of GEOS-Chem air quality model to

reproduce sharp $O_3$ vertical gradients in the first kilometre above surface of Southeast United-States (during both clear-sky and low-cloud conditions), attributed to excessive top-down mixing in the model. A thorough knowledge of the vertical distribution of pollutants within the PBL and at the interface with the FT is required for conducting diagnostic evaluations of CTMs through the entire PBL compartment, and not only at the surface. However, a common difficulty in the evaluation of CTMs relies in the possible error compensations that are often complex to identify. Indeed,

although closely linked, both PBL heights and pollutant concentrations (at the surface and/or along vertical profiles in the PBL) are often evaluated separately, which limits the significance of the drawn conclusions. For instance, a model may well reproduce the concentrations of a specific chemical compound at the surface but with an overestimated PBL and/or an overestimated vertical mixing, which would suggest that its sources are actually overestimated. A recent diagnostic evaluation of the WRF-Chem model focusing (for the first time) on the $O_3$ entrainment highlighted

deficiencies in the model, including an overestimation of the $O_3$ entrainment and a too efficient vertical mixing in the lower PBL (Kaser et al., 2017). These deficiencies were found to originate mainly from errors in the entrainment rate and PBL height during the morning, and an erroneous representation of the $O_3$ gradient at the PBL-FT interface during the rest of the day.

The general objective of this study is to derive a climatological vertical distribution of $O_3$ and CO over the period 1994-

2016 combined with information on the PBL. This paper focuses on these two pollutants but some results will also be (more briefly) discussed for the relative humidity (RH) and potential temperature ($\theta$). For this purpose, we benefit from the two main sources of in-situ vertical profiles in the troposphere : (i) the In-service Aircraft for a Global Observing System (IAGOS) database and (ii) the World Ozone and Ultraviolet Radiation Data Centre (WOUDC) ozonesondes database. We first implement an algorithm for automatic estimation of the PBL height from both sonde and airborne

profiles based on which we derive a climatological description of the vertical stratification of the $O_3$, CO, RH and $\theta$ within the PBL and at the interface with the FT. Many studies have already provided climatological vertical profiles of $O_3$ and CO, but most of the time simply by averaging individual profiles, no matter if the PBL height varies. The





original approach developed here consists of providing climatological vertical profiles in a vertical coordinate system based on the PBL height (hereafter referred to as PBL-referenced vertical profiles) : the altitudinal dimension of vertical profiles is first normalized by the PBL height, and then profiles are averaged to provide a climatological vertical distribution. While commonly used in studies dealing with PBL dynamics (e.g. Lilly, 2002), to our knowledge this approach has not yet been used to derive climatological vertical profiles of chemical compounds. The main benefit of this approach is to highlight possible specific features in the vertical distribution that would be smoothed with a simple average, in particular at the PBL-FT interface (an illustration is given latter in the text).

Data and methods for estimating PBL heights are described in Sect. 2. The climatological PBL heights are analysed in Sect. 3, and the vertical distributions in Sect. 4. The Sect. 5 presents a summary of the study and a discussion on the perspectives.

## 2    Data and methods

### 2.1    Data description

#### 2.1.1    MOZAIC-IAGOS observations

In the framework of the Measurements of OZone, water vapour, carbon monoxide and nitrogen oxides by Airbus In-service aircraft (MOZAIC) program and its successor the IAGOS program, observations of the chemical composition of the atmosphere have been routinely performed by commercial aircraft from several airline companies since 1994 for $O_3$ and RH and 2002 for CO (www.iagos.org) (Marenco et al., 1998; Petzold et al., 2015). Vertical profiles of the troposphere from the ground to about 9-12 km are obtained during the ascent and descent phases. Among the various parameters measured by IAGOS aircraft, we used in this study the barometric altitude, temperature, pressure, RH over liquid, $O_3$ and CO volume mixing ratios. The instruments and the period of data availability are summarized in Table 1. In both the MOZAIC and IAGOS programs, the same instrument technologies are used on all aircraft. During the 2009-2014 overlapping years, inter-comparisons have been systematically performed between MOZAIC and IAGOS, demonstrating a good consistency in the dataset (Nédélec et al., 2015). In MOZAIC, ozone was measured using a dual-beam UV-absorption monitor (time resolution of 4 seconds) with an accuracy estimated at about ±2 ppbv / ±2% (Thouret et al., 1998), while CO was measured by an improved infrared filter correlation instrument (time resolution of 30 seconds) with a precision estimated at ±5 ppbv / ±5% (Nédélec et al., 2003). In IAGOS, both compounds are measured with instruments based on the same technology used for MOZAIC, with the same estimated accuracy and the same data quality control. In MOZAIC, RH was measured by a compact airborne humidity sensing device using capacitive sensors (MOZAIC Capacitive Hygrometer MCH) (Helten et al., 1998; Neis et al., 2015a, 2015b; Smit et al., 2014). In IAGOS, RH is measured by the IAGOS Capacitive Hygrometer (ICH), a slightly modified version of the MCH (see Neis et al., 2015b for details). Instruments were calibrated for RH respect to liquid water. The absolute uncertainty on RH is estimated to ±5% RH. A more detailed description of the IAGOS system and its validation can be found in Nédélec et al. (2015). For convenience, the MOZAIC and IAGOS programs are hereafter commonly referred to as the IAGOS program. Although this study focuses on the period 1994-2016, it is worth noting that due to on-going calibration and validation of IAGOS data, all profiles are not yet available in a validated status after 2014. The ascent and descent rates of IAGOS aircraft are typically around 7-8 m s$^{-1}$ in the lower troposphere. Considering the time integration of the IAGOS instruments, this leads to a vertical resolution around 28-32 m for $O_3$ and RH and 210-240 m for CO.



### 2.1.2 Ozonesonde observations

In addition to IAGOS data, this study uses the ozonesonde observations over the period 1994-2016. These data are publicly available on the WOUDC database supported by Environment Canada (www.woudc.org), as part of the Global Atmospheric Watch (GAW) program of the World Meteorological Organization (WMO). Ozone is measured by three

main types of sensors (see Table 1) : Electro-chemical Concentration Cells (ECC) (~80% of the profiles), Brewer-Mast (BM) sensors (~10% of the profiles) and Carbon Iodine (CI) sensors (less than 10% of the profiles). The measurement uncertainties range from 3-5% with ECC to 5-10% with the other sensors (WMO, 2011). The response time of electrochemical cells of $O_3$ sondes typically ranges between 20 and 30 s (WMO, 2011), which gives an effective vertical resolution of 100-150 m for an ascent rate around 5 m s$^{-1}$. This is a factor 3-5 coarser than in IAGOS profiles.

The WOUDC profiles used in this study are performed with different types of radiosondes (e.g. Vaisala RS80 or RS92). The performance of the RH sensors deployed on these radiosondes depend on various factors (e.g. temperature, RH, solar radiation, altitude, presence of clouds) such that the overall uncertainties on RH are complex to quantify but can be estimated for the lower troposphere to be about 10% RH with respect to liquid water (Schröder et al., 2017).

### 2.1.3 Characteristics of airborne and sonde profiles

It is worth noting that the profiles obtained with balloons and in-service aircraft are intrinsically different. The horizontal displacement of the balloon throughout the PBL remains small, and the profile can thus be considered as vertical. Indeed, averaged between 0 and 4 km based on all ozonesondes available in 1994-2012, the mean ascent rate of ozonesondes is 5.6±0.9 m s$^{-1}$ (one standard deviation). It thus takes about 12 min for the balloon to reach 4 km of altitude. Considering a hypothetical wind of 10 m s$^{-1}$ in this layer, this would lead to an horizontal course displacement

of about 7 km. In comparison, the ascent/descent rates of IAGOS aircraft are faster (and more variable) : 7.3±2.0 m s$^{-1}$ on average (i.e. 9 min to reach 4 km). However, the aircraft horizontal speed is much stronger than the wind speed and increases with the altitude from about 85 m s$^{-1}$ at 0-1 km to 166 m s$^{-1}$ at 3-4 km on average. The horizontal displacement of IAGOS aircraft can thus be estimated to about 35 km to reach 2 km of altitude, and 70 km to reach 4 km. This issue has been discussed for the Frankfurt airport (where the number of available vertical profiles is the highest) in Petetin et

al. (2018). Therefore, the IAGOS profiles have to be considered as quasi-vertical profiles. At the scale of the FT, this is less problematic as the vertical variability is usually stronger than the horizontal one. However, it raises more questions within the PBL where the horizontal variability of meteorological and chemical parameters is stronger, especially in heterogeneous terrain/surface/environment. In order to assess how these differences influence the climatological vertical distribution of the chemical species and meteorological parameters, comparisons of the vertical distribution obtained

with IAGOS aircraft and ozonesondes taken separately will be provided in Sect. 4.

### 2.2 Data treatment

This study focuses on the northern mid-latitudes (25°N-60°N) in order to avoid ill-defined PBL in the tropics due to deep convection and sparse data in boreal and polar regions. In this region, IAGOS profiles are available at 135 airports and ozonesonde profiles at 20 stations, as shown in Fig. 1. The number of profiles available over the period 1994-2016

is 20,762 for the ozonesondes and 72,382 for IAGOS, which represents a total of 93,144 profiles. For both IAGOS and ozonesondes, most profiles are sampled in Europe and North America (especially in north-east Unites States), and a few in East Asia and Middle-East. It is worth noting at this stage that, as $O_3$ and CO mixing ratios can strongly vary from one location to the other, the PBL-referenced profiles that will be obtained from all profiles available at northern mid-latitudes are not expected to be representative of any location in this large latitudinal band. As profiles often show a

very complex structure, aggregating such a large number of profiles allows to smooth the vertical distribution and subsequently to highlight specific features. The idea of this study is to focus on the vertical stratification (i.e. the





relative changes of mixing ratios with altitude) of $O_3$ and CO rather than their mixing ratios themselves.

All profiles are expressed in meters above ground level (AGL). Note that the altitude available in the IAGOS database corresponds to the barometric altitude above sea level (ASL) estimated from the temperature and pressure measured by the aircraft, assuming standard conditions at the surface (temperature of 288.15 K, pressure of 1013.25 hPa). This leads

to an uncertainty on the actual altitude of the aircraft. Under some atmospheric conditions (cyclonic conditions for instance), the barometric altitude of the aircraft may be below the airport elevation. Without any information on the temperature and pressure at the surface close to the airport, it is not possible to get a more accurate estimation of the altitude. In this study, the altitude AGL is deduced from the barometric altitude ASL available in the IAGOS database by subtracting its first value measured by the aircraft, assuming that this first measurement of the profile is performed

close to the surface. The IAGOS measurements are indeed programmed to start when the aircraft wheels leave (or touch) the ground. Some technical issues delaying the beginning of the measurements may occur, but this is expected to concern only a minor proportion of the profiles. Note that the GPS altitude is available in the IAGOS database only since 2014.

For convenience, all profiles are linearly interpolated at a vertical resolution of 50 m from the surface (thus, this value

of 50 m is to be considered as the truncation error in our study). This value was chosen following the sensitivity analysis on the vertical resolution (from 1 to 10 hPa, i.e. about 10 to 100 m) recently performed by Liu and Liang (2010), who concluded that it represents a good compromise between accuracy and uncertainty related to data noise. A sensitivity test with a vertical resolution of 100 m (not shown) confirmed the low sensitivity of results to this parameter. Additionally, at any given (50 m-deep) level, no interpolation is performed when the vertical distance between the two

neighbouring points of the raw vertical profile used for the interpolation exceeds 100 m. In this case, the data is considered as missing.

As the PBL characteristics depict strong diurnal variations, profiles used in this study are separated into different time slots : night (sunset to sunrise), morning (sunrise to solar noon), midday (solar noon to 3 h past solar noon), afternoon (3 h past solar noon to sunset), daytime (sunrise to sunset) and the whole day (denoted "all" in the figures). All time zones

and daylight saving hours are properly taken into account.

### 2.3    Estimation of PBL heights

Three types of PBL can be distinguished : the convective boundary layer (CBL) often occurring during daytime and characterized by a strong turbulent mixing under the effect of convective thermals, the stable boundary layer (SBL) occurring mainly during nighttime and characterized by the absence of turbulence mixing, and the residual layer (RL)

occurring mostly during the night and morning and corresponding to the former CBL and usually delimitated by the SBL top and the capping inversion (Stull, 1988). In our study, for all individual profiles, we first look for any surface-based inversion (SBI) of temperature defined as a monotonic increase of (absolute) temperature from the surface up to a certain altitude (corresponding to the top of the SBI). When no SBI is found, numerous methods have been proposed over the past decades for estimating the PBL height (see Seibert et al., 2000, for a review), including : (i) the elevated

inversion (EI) method in which the PBL top is located at the bottom of an elevated (absolute) temperature inversion, (ii) the methods based on the search for an extremum of vertical gradient in the vertical profile of a relevant thermodynamic parameter (e.g. RH, potential temperature, refractivity), and (iii) the methods in which the profile is scanned upward in order to identify at which altitude a certain thermodynamic parameter (e.g. virtual potential temperature, bulk Richardson number) equalises or exceeds by a certain amount its surface value. Strong systematic differences of PBL

height are found among these methods, both in terms of magnitude and seasonal/diurnal variability (e.g., Seidel et al., 2010; Wang and Wang, 2014). The reasons for the discrepancies between the methods are complex (and not clearly understood yet), but may comprise a poor vertical mixing of the PBL, the strong influence of the surface measurement





(specifically for the last class of methods), the existence of clouds, the uncertainties on the RH measurements under cloudy conditions (Wang and Wang, 2014). As no consensus currently exists, we decided to retain the EI approach in which the PBL top is estimated as the first altitude above which the (absolute) temperature monotonically increases with altitude. The vertical gradient of temperature between the top and the bottom of the EI corresponds to the intensity

of the inversion. We require that the difference of temperature between the EI base and the altitude level right above exceeds the value of 0.3 K in order to avoid erroneous identification of the EI base due to uncertainties on the temperature measurements estimated at ±0.25 K in IAGOS (Berkes et al., 2017). All profiles with no or too weak (below 0.3 K) temperature inversion are discarded. Two examples of profiles are presented in Fig. 2. Note that as it relies only on temperature and not on RH measurements (that are not always fully available over the profiles since all

RH data flagged as "doubtful" in IAGOS are rejected), the EI method allows to maximize the number of profiles then taken into account for deriving the climatological vertical distribution of $O_3$ and CO.

Although the EI represents a real geophysical interface between two layers, it is important to note at this stage that this height does not necessarily always correspond to the height of the mixing layer as it may for instance correspond to the capping inversion aloft the residual layer (rather than e.g. the top of an instable nocturnal boundary layer or a growing

CBL in the morning). For convenience, we will hereafter refer to the PBL height but the reader should keep in mind that this term may sometimes be ambiguous.

Following previous studies (e.g., Seidel et al., 2010; Wang and Wang, 2014), a maximum PBL height of 4,000 m AGL is fixed. In order to further avoid erroneous PBL height estimations due to too large data gaps in the profile, we require at least 75% of available data between 0 and 4,000 m AGL. In addition, for all PBL calculations, we require a

maximum of 200 m (i.e. four 50 m-deep levels) with missing data between the surface and the estimated PBL height. Among the 93,144 profiles available, SBI and EIs are found on 16% and 63% of the profiles, respectively. The remaining profiles (21%) either do not fulfil the previous criteria (due to data gaps) and/or show no significative temperature inversion in the first 4,000 m AGL, and are thus discarded.

## 3   PBL height results

In this section, we analyse the climatology of the PBL heights obtained from the IAGOS and ozonesonde profiles by distinguishing the case of SBIs (Sect. 3.1) and EIs (Sect. 3.2).

### 3.1   Surface-based inversions (SBIs)

SBIs are important features for air pollution as they inhibit the vertical mixing of pollutants released at the surface. Concerning ozone, they can induce a strong depletion either by dry deposition or titration by the nitrogen oxide (NO)

accumulated at the surface (Colbeck and Harrison, 1985). The distribution of the local time (LT) at which (IAGOS and ozonesondes) profiles are measured is shown in Fig. 3 with the proportion of SBIs occurrences. Most of the available profiles are measured between 05:00 and 19:00 LT. Results highlight a strong diurnal variability of these SBIs and their characteristics. As expected, they are the most frequent during the night when their proportion continuously increases up to a maximum of 60% at 03:00 LT (red curve in Fig. 3). Thus, although SBIs are more frequent during the night,

many night-time profiles are still showing unstable conditions. At the locations close to large agglomerations (e.g. airports), the absence of SBIs during the night may be partly due to the urban heat island phenomenon that can turn a stable PBL into a near-neutral PBL (Dupont et al., 1999). The proportion of SBIs then progressively decreases to a broad minimum of 5-10% between 10:00 and 17:00 LT. Very similar diurnal variations of the proportion of SBIs are observed during all four seasons (not shown).



Both the height and the temperature lapse rate (vertical gradient of temperature between the surface and the SBI top) of these SBIs are shown with their diurnal and seasonal variations in Fig. 4 (considering both IAGOS and ozonesondes profiles). SBIs occur all along the year with almost no seasonal variations in their frequency. This type of inversion leads to very shallow stable layers with a mean depth of 110 m. A moderate seasonal variability of 23% is highlighted

(as calculated as the difference between the maximum and the minimum SBI height normalized by the annual SBI height), with mean SBI heights ranging from 97 m in summer to 123 m in winter. However, the diurnal variability of the SBI height is strong (71%), the largest SBIs being observed during night-time (131 m on annual average) and the smallest during mid-day (53 m on annual average). The 95[th] percentile reaches about 300 m. This is substantially lower than the SBIs reported by Seidel et al. (2010) whose median heights were ranging around 200-500 m. Differences may

be (at least partly) due to the fact that our results are based on profiles measured at northern mid-latitudes stations (most of them being located in the latitudinal band 35°N-55°N) essentially during daytime, while the dataset analysed by Seidel et al. (2010) extends to stations located further North up to polar regions and includes a predominant proportion of nighttime/morning radiosonde profiles. On average, the temperature lapse rate is about 13 K km$^{-1}$, with moderate seasonal differences (12%). Some diurnal variations are also observed during most seasons, with usually decreasing

temperature lapse rates from night-time to the afternoon.

### 3.2    Elevated temperature inversions (EIs)

Some characteristics (temperature difference, width and temperature vertical gradient) of the EIs are shown in Fig. 5. The difference of temperature between the base and the top the EIs is 1.5 K on annual average, with strong seasonal variations from 1.1 K in summer to 2.1 K in winter. The 95[th] percentile also depicts a high seasonality with values

ranging from 3.3 K in summer to 7.2 K in winter. The mean width of these EI increases from 76 m in summer to 103 m in winter, in reasonable agreement with EI thicknesses estimated by other authors (e.g., Cohn and Angevine, 2000). This leads to mean temperature vertical gradients of 1.4 and 1.9 K hm$^{-1}$ during these two seasons, respectively. Interestingly, none of these characteristics depicts a diurnal variation (whatever the statistical metric).

The seasonal and diurnal variations of the PBL height estimated with the EI method are shown in Fig. 6 for all seasons

and time slots. Averaged over all profiles, the mean PBL height is 1,253 m, with values ranging from 1,132 m during the night to 1,483 m in the afternoon. This corresponds to a diurnal variability of 28%, this diurnal variability being here calculated as the maximum minus minimum PBL height normalized the mean PBL height, based on the values available during at the different time slots shown in Fig. 6 (for this example: (1,483-1,132)*100/1,253 = 28%). Such high nighttime values and moderate diurnal variability indicate that the EI height often corresponds to the top of the

nocturnal RL (as previously mentioned in Sect. 2.3). As expected, the diurnal variability is much lower in winter (17%) than in the other seasons (30-38%). The highest PBL heights are observed during summertime afternoon with 1,707 m on average. Similarly, the seasonal variability strongly varies with the time of day, with values of 22, 23, 32 and 35% during the night, morning, midday and afternoon, respectively.

### 4    PBL-referenced vertical distribution

In this section, we investigate the climatological vertical stratification of two thermodynamic parameters (RH, θ) and two trace gases (O$_3$, CO) within the PBL (estimated with the EI method) and at the PBL-FT interface. Due to the variations of PBL height from one profile to the other, calculating a climatological profile by simply averaging all individual profiles inevitably smoothes all the vertical features that may exist for some compounds or meteorological parameters, especially at the PBL-FT interface. In order to highlight how the PBL height influences these vertical

distributions, all individual vertical profiles are thus first expressed in a vertical coordinate system based on the PBL



height and then averaged. In practice, all individual profiles are expressed as a function of $z/h$ with $z$ the altitude and $h$ the PBL height estimate, with $z/h$ ranging from 0 to 2 (in bins of 0.05). For instance, if the PBL height on a specific profile is 1,000 m, the re-sampled profile will extend from 0 to 2,000 m (with bins of 50 m). Hereafter, this type of vertical profile is denominated as a PBL-referenced vertical profile. Then, all these PBL-referenced profiles are

averaged to derive a climatological vertical distribution apart from the PBL-FT interface. Hereafter, the PBL-FT interface will designate the $z/h=1$ altitude level (which means that the entrainment zone is here included in the FT).

In order to illustrate the usefulness of this approach, we consider an artificial dataset of vertical profiles characterized by the presence of a discontinuity at the PBL top ($h$) here arbitrarily chosen as a step function with different but constant mixing ratios below ($c_1$) and above ($c_2$) this interface. A dataset of 50 profiles is generated by choosing random integers

between 10 and 30 ppbv for $c_1$, between 20 and 40 ppbv for $c_2$, and between 100 and 1,500 m for $h$. All these profiles are superposed in grey in Fig. 7 (top left panel), including one individual example in black (for which $c_1$=15 ppbv, $c_2$=38 ppbv and $h$=1,200 m). In such a dataset, the mean vertical distribution (red curve in top left panel) is characterized by an overall increase of the mixing ratios with altitude (up to 1,500 m, the upper bound fixed in this example). The discontinuity introduced in all individual profiles is entirely smoothed in this mean profile, as shown by

the gradient profile (red curve in top right panel). If all these profiles are first normalized by the PBL height (grey curves in bottom left panel) and then averaged (blue curve), the discontinuity of the mixing ratios is preserved, as clearly shown by the PBL-referenced gradient profile (bottom right panel). For a comparison with the traditional profiles, both PBL-referenced profile and gradient profile were dilated using the mean PBL of this dataset (about 900 m in this example) and added to the first series of plots (blue curves in top panels). Considering PBL-reference profiles

thus allows to investigate the features (or in other words, any kind of discontinuity) that may exist at the PBL top.

Only complete profiles (i.e. with available data at all $z/h$ from 0 to 2) are averaged together. This is an important restriction which limits the number of profiles but ensures the most reliable vertical distribution when averaged. Note that we tested to fill the small data gaps (width up to 100 m included) by interpolation using natural cubic splines. As this only increases the number of profiles by about 10% (which means that the data gaps are usually larger than 100 m)

and does not change the climatological results, we decided to not use any interpolation to fill the data gaps in the profiles. The number of complete profiles is finally 43,244 for the potential temperature, 17,649 for RH, 30,960 for $O_3$ and 8,295 for CO.

In the following sections, the PBL-references profiles are presented for θ (Sect. 4.1), RH (Sect. 4.2), CO (Sect. 4.3) and $O_3$ (Sect. 4.4) based on the profiles on which an EI is identified. Mean profiles will be shown for the different seasons

and time slots. However, it is worth noting that the climatological profiles at the different time slots are not directly comparable between each other since they are calculated based on profiles sampled at different periods and locations. The profiles of (local) vertical gradients will be also analysed. Note that for all variables discussed in Sect. 4 (θ, RH or mixing ratios), these profiles of vertical gradients will be expressed in the common unit of the variable (°C, % or ppbv) per hectometre ($hm^{-1}$) as it keeps most numbers in the range of 0.1-10.

### 35    4.1    Potential temperature

The PBL-referenced profiles of potential temperature in the absence of SBI (and in the presence of an EI) are shown in Fig. 8. On average, the potential temperature is found to be slightly superadiabatic in the surface layer (i.e. θ decreases with altitude) during both morning (-0.08°C $hm^{-1}$) and midday (-0.13°C $hm^{-1}$). The width of this superadiabatic layer never exceeds 5-10% of the PBL height.

Above that surface layer, the potential temperature increases with altitude. While neutral adiabatic profiles (i.e. no variations of θ with altitude) were expected within the convective PBL, at least during daytime (Stull, 1988), all climatological profiles appears subadiabatic (positive vertical gradient of θ). This subadiabatism varies with the season



with strongest values in spring/winter (0.51°C hm$^{-1}$ on averaged over the PBL) and lowest values in summer (0.25°C hm$^{-1}$). As expected, a very sharp increase of the potential temperature is highlighted at the top of the PBL where vertical gradients reach +1.3°C hm$^{-1}$ on average. This maximum gradient is found to be much higher in winter (+1.6°C hm$^{-1}$) than in summer (+1.1°C hm$^{-1}$), as previously analysed (see Sect. 3.2). These seasonal variations are the strongest during

the afternoon (+1.8 and +1.0°C hm$^{-1}$ in winter and summer, respectively). Above, in the lower FT, the increase of temperature with altitude is reduced but remains higher than in the PBL.

As mentioned in Sect. 2.1, several important differences of sampling exist between these two datasets, including that (i) IAGOS aircraft fly at a 30% higher descent/ascent rate and cover a horizontal distance 10 times larger in the lower troposphere, (ii) IAGOS measurements are performed in the vicinity of international airports and large agglomerations

while ozonesondes stations are usually located in remote, rural or low-density urban area (leading to a population density around IAGOS airports of about 2,000 inhabitants km$^{-2}$ on average within ±0.1° in longitude and latitude, against about 1,150 inhabitants km$^{-2}$ for ozonesonde stations; see Sect. S1 in the Supplement for details). This raises the question of whether or not these sampling differences impact the PBL-referenced vertical distribution. Answering this question would require collocated (in time and space) IAGOS and ozonesonde profiles, but far too few profiles fulfil

these conditions. However, in order to give some insights about this question, we calculated the climatological profiles from both datasets taken separately (Fig. 9). Only daytime profiles are shown as ozonesondes are much sparser during the night. Again, these PBL-referenced profiles obtained with IAGOS and ozonesondes are not expected to be the same since they are sampled in different locations and times and thus correspond to different PBL heights; considering the profiles used in Fig. 9, the mean PBL height is 10-20% higher in ozonesondes than in IAGOS profiles depending on the

season.

Results obtained using both datasets are in reasonable agreement. The main difference is found near the surface where only ozonesondes highlight a small superadiabatism. This may be partly due to the fact that contrary to ozonesondes, IAGOS measurements do not start at the surface but at a minimum height of a few meters AGL (since instruments are located in the lower part of the fuselage). However, as individual IAGOS profiles sometimes do show a superadiabatism

at the surface, the main reason is more likely related to the inherent uncertainties of the IAGOS barometric altitude which is deduced from the pressure assuming standard atmospheric conditions at the surface (as explained in Sect. 2.2). This may partly smooth the superadiabatism existing at the surface (as the mean potential temperature in the first 0-50 m altitude level would include some points actually outside the 0-50 m layer and/or ignore some other points actually belonging to this layer). At the annual scale, considering all time slots and $z/h$ levels, the mean bias (MB), root-mean

square error (RMSE) and correlation (R) between the PBL-referenced profiles of both datasets are 2°C, 3°C and 0.97, respectively (with ozonesondes here taken as the reference).

### 4.2     Relative humidity

The PBL-referenced profiles of RH are shown in Fig. 10. At the surface, the mean RH ranges between 55 and 80% with a well-know seasonal and diurnal pattern characterized by highest values during the wintertime nights and lowest values

during the springtime/summertime afternoons. As one moves higher in altitude, RH increases quite regularly up to a maximum located around $z/h$=0.8, thus just below the top of the PBL. At this level, RH values range between 70 and 85%. The seasonal differences persist but the diurnal ones are greatly reduced (the absolute difference between the nighttime maximum and the afternoon minimum remains below 10 %). A sharp decrease of RH is observed at the PBL-FT interface. The vertical gradient reaches its minimum right above the PBL top (at $z/h$=1.05) with -12% hm$^{-1}$ on

average. The diurnal and seasonal variability of these strongest RH gradients remains low (between -11 and -15% hm$^{-1}$). In the lower FT, RH decreases with altitude, usually with stronger (negative) gradients in winter than in summer.





The PBL-referenced vertical profiles of RH obtained with IAGOS and ozonesondes taken separately are shown in Fig. 11. Although the shape of the profiles remains in reasonable agreement, some differences are highlighted. In particular, ozonesondes show a stronger RH vertical gradient within the PBL and a sharper decrease in the lowermost FT with much lower RH in the FT. This sharper decrease above the PBL top might be due to the fact that, as briefly mentioned

in Sect. 2.1.2, RH measurements with radiosondes are generally affected by a radiation dry bias due to the heating of the sensors by the solar radiation, which can lead to a negative bias on the RH measurements on the order of 5-10% in the lower troposphere (e.g. Vömel et al., 2007; Bian et al., 2011; Wang et al., 2013). In our case, this bias could be further enhanced in the lower FT due to solar reflection by clouds at the top of the PBL. This is also supported by the fact the differences between IAGOS and sondes are largely reduced when considering only nighttime profiles, i.e. when

radiosonde measurements are not affected by heating effects due to solar radiation (not shown). At the annual scale, taking into account all time slots and $z/h$ levels and considering ozonesondes as the reference, the comparison between both IAGOS and ozonesonde datasets gives MB, RMSE and R of +0%, 9% and 0.67, respectively.

### 4.3    Carbon monoxide

The PBL-referenced profiles of CO are shown in Fig. 12. The uncertainties of these mean profiles are substantially

higher than for the previous meteorological parameters notably due to a much lower number of measurements (as CO is only measured by IAGOS aircraft and starting from 2002). Considering all profiles, the mean CO mixing ratios at the surface increase from about 240 ppbv in summer to 340 ppbv in winter. The CO mixing ratios decrease with altitude at a varying rate depending on the altitude. The normalized profiles (middle panels in Fig. 12) show that the difference of CO between the surface and the PBL top reaches a factor 1.3 on average. The first important result shown by these

PBL-referenced profiles is therefore the substantial vertical stratification of CO mixing ratios within the PBL.

In order to investigate that stratification on a quantitative basis, we introduce a first factor of vertical stratification (γ) here defined as the standard deviation of the mixing ratio profile between the surface and the PBL top (i.e. over the 21 $z/h$ levels comprised between 0 to 1) normalized by the mean. We also define a second factor of vertical stratification (δ) calculated by normalizing γ by the PBL height. The calculations are first done on each individual profiles and then

averaged over all profiles. The γ and δ factors will be expressed in % and % hm$^{-1}$, respectively. Results are reported in Table 2 for the different seasons and time slots. On average, the γ factor of CO vertical stratification is 11% and varies little with the season and time of day (from 8 to 12%), the strongest values being usually found in winter/fall. The δ factor depicts relatively stronger variations, with values ranging from ~1% hm$^{-1}$ during summer midday to more than ~2% hm$^{-1}$ during winter/fall night.

The strongest vertical gradients are observed at the PBL top and close to the surface. At the PBL top, the strong vertical gradients involve the presence of a clear inflexion point in the mean CO profile (as shown by gradient profiles in the right panels of Fig. 12). With a traditional vertical coordinate system (i.e. $z$ rather than $z/h$), this features would have been smoothed (see Fig. 7). The presence of this inflexion point on an independent variable (i.e. a variable not used in the estimation of the PBL height) gives confidence on our ability to capture reasonably well a real geophysical interface

with the EI approach. It illustrates the fact that as expected, EIs act as an effective although porous geophysical barrier that limits the vertical exchanges between PBL and FT, leading to distinct chemical composition on each side. The sharp gradients at the PBL-FT interface are strongest in winter and lowest in summer. Such seasonal variations are consistent with the fact that EIs are deeper and characterized by a stronger temperature gradient in winter than in summer (as previously shown in Sect. 3.2), which greatly inhibits the ventilation of the polluted PBL and the exchanges

with the cleaner FT. The strong gradients at the surface ensue from the presence of CO emissions and are usually maximum during nighttime and morning. Substantially lower vertical gradients are found in the FT.



### 4.4    Ozone

#### 4.4.1    PBL-referenced vertical distribution of O₃

Figure 13 presents the mean PBL-referenced profiles of $O_3$ in which all profiles sampled at northern mid-latitude stations are aggregated. It is worth noting that both surface mixing ratio and vertical distribution of $O_3$ are highly
variable in both time and space and can greatly change depending on the meteorological conditions and the availability of $O_3$ precursors. At the scale of individual profiles, the $O_3$ vertical distribution is often very complex with persistent layering. In our study, taking into account a very large number of profiles allows to obtain well smoothed PBL-referenced profiles (Fig. 13) with low layering and therefore to highlight some general background characteristics of the $O_3$ vertical stratification in the lower troposphere.

On average over all profiles, surface $O_3$ mixing ratios range between 20 ppbv in winter/fall to 30-35 ppbv in spring/summer. Above the surface, the $O_3$ mixing ratios increase with altitude through the whole PBL and the lower FT with vertical gradients depicting strong variations depending on the season and altitude (relatively to the PBL top). As for CO, the strongest gradients are observed both close to the surface and at the PBL-FT interface. Close to the surface, they are likely explained by the strong intensity of the main $O_3$ sinks, namely dry deposition and titration by the NO
emitted by anthropogenic emission sources. The combination of these two sinks leads to sharper vertical gradients for $O_3$ than for CO (relatively to mixing ratios). The maximum vertical gradient at surface is observed during the night (around 3 ppbv hm$^{-1}$). The rate of $O_3$ increase with altitude slightly decreases with altitude in the PBL. A clear inflexion point is highlighted at the interface between PBL and FT ($z/h$=1). Compared to CO, this inflexion point in $O_3$ profiles is usually much sharper. Such a difference suggests that, (i) while the smooth CO inflexion point mostly results from the
limited vertical exchanges between PBL and FT in the presence of the EI, (ii) the stronger $O_3$ inflexion point is not only due to this dynamical effect but also to a difference of chemical regime apart from the PBL-FT interface. In other words, results suggest that the CO inflexion point is mostly driven by dynamics (since the CO chemical reactivity is low) while the $O_3$ inflexion point is driven by both dynamics and chemistry. The vertical gradient at the PBL top is found to strongly vary with the season with the sharpest increase in winter and a much smoother increase in summer.
While this strong gradient persists all day along in winter, it is found to be lower in midday/afternoon during summertime. This is likely due to the combined effect of an efficient photochemical production of $O_3$ in the PBL and the entrainment of $O_3$-rich air masses from the FT. The entrainment can indeed play a strong role in the $O_3$ budget within the PBL, comparable to advection or deposition, as recently highlighted by Trousdell et al. (2016). Higher in altitude, in the lower FT, the increase of $O_3$ mixing ratio with altitude is found to be substantially smaller than in the
PBL.

Based on 214 aircraft vertical profiles obtained during the DISCOVER-AQ (Deriving Information on Surface conditions from Column and Vertically Resolved Observations Relevant to Air Quality) and the FRAPPÉ (Front Range Air Pollution and Photochemistry Éxperiment) campaigns in Colorado during summer 2014, Kaser et al. (2017) recently investigated the $O_3$ vertical gradient between the PBL and the lower FT in order to estimate this $O_3$ entrainment
in the PBL and to evaluate its representation in the WRF-Chem model. The difference of $O_3$ mixing ratio between the PBL and the (arbitrary chosen) 300 m-wide layer above the PBL top was found to vary from +9 ppbv in the morning to -11 ppbv in the afternoon (the negative value meaning that higher $O_3$ mixing ratios are measured in the PBL). This differs from our climatological results in which the summertime $O_3$ vertical gradients at the PBL-FT interface are also decreasing from the morning (2.5 ppbv hm$^{-1}$) to the afternoon (0.9 ppbv hm$^{-1}$) but remain positive (i.e. $O_3$ mixing ratios
in the FT remain higher than in the PBL). However, if we consider only the ozonesonde profiles at Boulder, Colorado (i.e. in the same region where the DISCOVER-AQ campaign took place), our results show a summertime vertical gradient of $O_3$ at the PBL-FT interface close to zero during the late morning and negative at midday (see Fig. S-1 in the





Supplement), thus in good agreement with Kaser et al. (2017). Our study also agrees with Kaser et al. (2017) on the fact that, even at daytime during summer (when the vertical turbulent mixing is expected to be the strongest), a strong vertical stratification of $O_3$ mixing ratios persists in the lower part of the PBL (the first few 100 m).

As mentioned at the beginning of this section, comparing the climatological profiles at the different time slots is tricky since they are partly based on profiles sampled at different locations. However, in terms of diurnal variations of $O_3$, the 18 years of IAGOS profiles available at the Frankfurt airport have already been used to demonstrate the clear decrease of the diurnal variability with altitude (Petetin et al., 2016).

In terms of vertical stratification within the PBL, the $\gamma$ and $\delta$ factors for $O_3$ are given in Table 2. Considering all profiles, the mean vertical stratification of $O_3$ in the PBL is 18% (or 5% $hm^{-1}$). It ranges from ~10% (~1% $hm^{-1}$) in spring/summer midday/afternoon to ~25% (7-10% $hm^{-1}$) in winter/fall night. This is consistent with a stronger vertical mixing within the PBL associated to a higher thermal instability in the PBL under sunny conditions. In order to investigate the influence of meteorological conditions, both stratification factors are calculated for different ranges of surface potential temperature (from -10 to +35°C, with bins of 5°C) considering only daytime profiles (Fig. 14, top panels). For comparison, results are also shown for CO (bottom panels). Note that whatever the season, computing the weighted average of the curve shown in Fig. 14 with weights taken as the number of profiles available at each potential temperature interval allows to retrieve the mean (daytime) $\gamma$ factors given in Table 2.

Contrary to CO vertical stratification factors that do not show clear variations with the surface potential temperature, $O_3$ results highlight an interesting bell shape. Weakest $O_3$ vertical stratification is observed not only at high potential temperatures (above 30°C, when turbulence is expected to be strong due to the heating of the surface), but also at low (typically negative) potential temperatures, while strongest stratification is observed at intermediate potential temperatures. This behaviour is observed during all seasons (except summer when temperatures remain high enough). During wintertime, a relatively well-mixed $O_3$ profile with moderate mixing ratios is for instance frequently observed at stations in northern North America (e.g. Goose Bay, Yarmouth, Churchill). Note that this decrease of the vertical stratification is not due to much lower $O_3$ mixing ratios at the surface (the denominator in $\gamma$'s and $\delta$'s formula) since the latter also follows an (inversed) bell shape with a decrease from 23 to 17 ppbv between -10 and 0°C and an increase up to 48 ppbv at 30°C (see the density scatter plot of $O_3$ mixing ratios versus surface potential temperature in Fig. S-2 in Supplement). Similarly, low standard deviations in $O_3$ profiles within the PBL (the numerator in $\gamma$'s and $\delta$'s formula) are usually found when potential temperatures are negative. The weaker vertical stratification at lower surface potential temperatures illustrates the fact that although this last parameter alone could be somehow a relevant proxy of the static instability within the PBL, the vertical homogeneity of the $O_3$ pollution in this layer is also driven by other mechanisms (e.g. wind, clouds, snow), especially under cold conditions. Concerning the influence of snow, although still very uncertain, much lower $O_3$ deposition rates have been reported in the literature over snow compared to vegetation, due to the low reactivity of $O_3$ in pure water (e.g. Helmig et al., 2007; Stocker et al., 1995; Wesely et al., 1981). This could at least partly explain the weaker $O_3$ vertical stratification under low negative surface potential temperatures (while such bell shapes are not observed with CO for which no deposition sink exists), although dedicated studies are obviously required for investigating in more details the reasons for such a behaviour.

### 4.4.2 Comparison between IAGOS and ozonesondes profiles

As for the previous meteorological parameters and chemical species, we now investigate how PBL-referenced profiles obtained from IAGOS and ozonesondes are comparable. The climatological profiles from both datasets taken separately are compared in Fig. 15, considering only daytime profiles. As expected, some quantitative differences of $O_3$ mixing ratios are found between both datasets likely due to their different spatio-temporal distribution at the northern mid-latitudes (IAGOS showing lower mixing ratios). However, the normalized profiles highlight a consistent vertical



structure of $O_3$ between IAGOS and ozonesondes, whatever the season. One difference is the slightly less pronounced decrease of $O_3$ right above the EI base in ozonesondes profiles. This could be due to their longer sensor response time of ozonesondes (20-30 s against 4 s for IAGOS) and the subsequent coarser vertical resolution of the sampling that smoothes more the sharp vertical gradients in the capping inversion layer.

### 4.4.3    Vertical autocorrelation

In this section, we analyse the vertical autocorrelation of $O_3$ mixing ratios in the $z/h$ vertical coordinate system, in order to further investigate the links between the PBL and the FT. The vertical autocorrelation designates the correlation of mixing ratios between two different altitude levels. Based on all individual profiles, we calculate the correlation (R) between the different pairs of $z/h$ altitude level. The obtained $O_3$ vertical autocorrelation matrix is shown in Fig. 16. The CO matrix is also shown for comparison. Results highlight a difference of variability apart from the PBL-FT interface. Indeed, within both PBL ($z/h$ between 0-1) and FT ($z/h$ between 1-2), strong correlations are found, usually above 0.75. Conversely, correlations between the two atmospheric compartments are found to decrease more quickly with the vertical distance, as illustrated by the ("wave") shape of the iso-correlation contours. For instance, $O_3$ mixing ratios at $z/h$=0.9 (i.e. just below the PBL top) appear highly correlated with $O_3$ mixing ratios in the entire PBL with correlations above 0.75 down to $z/h$=0.0.5, but correlations are found to (more) quickly deteriorate with altitude in the FT, the 0.75 threshold being reached at $z/h$=1.4. Similar results are found for CO, except that the change of correlation apart from the PBL-FT interface is slightly more smoothed compared to $O_3$. This would be consistent with the fact that the differences of CO between PBL and FT are mostly explained by the dynamics (transport), contrary to $O_3$ for which the chemistry prevailing within the PBL helps to differentiate more strongly the $O_3$ mixing ratios and their variability in the two atmospheric compartments (as discussed in Sect. 4.4.1).

## 5    Summary and conclusion

In this study, we investigated the vertical stratification of $O_3$ and CO in the PBL and at the interface with the FT. Some results were also given for the potential temperature and RH. We collected all the in situ vertical profiles performed by WOUDC ozonesondes and IAGOS aircraft at the northern mid-latitudes (Fig. 1). Over the period 1994-2016, this represents a dataset of more than 90,000 profiles (78% of IAGOS profiles, 22% of sondes profiles).

As a preliminary step, we used all temperature profiles to identify surface-based and elevated temperature inversions (denoted SBIs and EIs, respectively). The occurrence of SBIs was found to strongly vary along the day, with frequencies ranging between 10% at midday and 60% in very early morning (Fig. 3). Results also highlighted strong diurnal variations of the characteristics of SBIs, the deepest and strongest SBIs being observed during the night (Fig. 4). However, no particular seasonal variation of SBIs was observed. On the profiles without SBIs, we looked for EIs, the base of which was taken as an estimate of the PBL height. This approach allows to identify where the capping inversion occurs but this likely does not always correspond to the PBL height as it may sometimes correspond to the top of a residual layer (especially during the night, or in the morning when the PBL is not fully developed). Contrary to SBIs, EIs depicted no diurnal variations but some weak seasonal variations, with the deepest (thinnest) and sharpest (smoothest) EIs occurring during winter (summer); the strength being here represented by the temperature lapse rate within the inversion layer (Fig. 5). The climatological PBL heights as determined with the EI method were found to be consistent with the results obtained by Seidel et al. (2010) through a more exhaustive analysis of meteorological sondes (Fig. 6).




Based on these PBL height estimations (denoted $h$), we built the so-called PBL-referenced vertical distribution of RH, $O_3$ and CO, calculated by averaging all individual profiles formerly expressed as a function of $z/h$ (with $z$ the altitude). Considering $z/h$ rather than $z$ aims at shedding light on the features existing at the PBL-FT interface which would have been smoothed otherwise (Fig. 7). For all meteorological parameters (Fig. 8 and 10) and chemical species (Fig. 12 and 13), the PBL-referenced profiles highlighted clear inflexion points at the PBL top, which gives support to our ability to capture reasonably well a real geophysical interface with the EI method. Comparing the PBL-referenced profiles obtained with IAGOS and ozonesondes taken separately showed a broad consistency for potential temperature and RH, although some differences exist (Fig. 9 and 11). In order to quantify how well pollutants are mixed within the PBL, we introduced two factors of vertical stratification, the first ($\gamma$) being defined as the standard deviation of the profile in the PBL ($z/h$ between 0 and 1) normalized by the mean, the second ($\delta$) being defined as $\gamma$ normalized by the PBL height (Table 2). Results showed that the frequently assumed well-mixed PBL remains an exception. The $\gamma$ ($\delta$) vertical stratification of CO within the PBL was 11% (1.7% hm$^{-1}$) on average, with some seasonal and diurnal variations (only for $\delta$). A stronger vertical stratification was found for $O_3$ with a $\gamma$ ($\delta$) factor of 18% (5.1% hm$^{-1}$) on average. The seasonal and diurnal variations of the $O_3$ vertical stratification were also stronger than for CO, with values ranging from ~10% in spring/summer midday/afternoon and ~25% in winter/fall night. For both species, this vertical stratification was not uniform through the PBL as stronger vertical gradients were found at both the surface (dry deposition and titration by NO for $O_3$; surface emissions for CO) and the PBL-FT interface. These vertical gradients at the PBL top strongly vary with the season, with maximum (minimum) values in winter (summer). In comparison, lower vertical gradients were found in the lower FT. Investigating the variations of the vertical stratification factors with the surface potential temperature highlighted an interesting bell shape for $O_3$ (but not for CO) with weakest stratification at both lowest (typically negative) and highest temperatures (Fig. 14). This could be due to a substantial decrease of the $O_3$ dry deposition under the presence of snow, although dedicated studies are required to confirm or infirm this hypothesis. Consistent PBL-referenced profiles were obtained for IAGOS and ozonesondes taken separately (Fig. 15).

Therefore, these results illustrate the fact that EIs indeed act as a geophysical interface between the PBL and FT. Compared to CO, the $O_3$ PBL-referenced profiles depict a sharper inflexion point at the PBL-FT interface, which suggests that the CO inflexion point may mostly due to dynamics (since its chemical reactivity is low), while the stronger $O_3$ inflexion point would result from the combined effect of both dynamics and chemistry (different chemical regimes between PBL and FT). This is also supported by the matrices of vertical autocorrelation that highlighted lower correlations apart from the PBL-FT interface and higher correlations within each of the two atmospheric compartments (PBL and FT), especially for $O_3$ (Fig. 16).

This study focused on the general characteristics of the $O_3$ and CO vertical stratification at northern mid-latitudes by aggregating the largest amount of profiles. It would be interesting in the near-future to investigate how these PBL-referenced profiles differs depending on the environment (urban, rural, coastal, remote) or eventually the region. In order to be able to make some relevant comparisons, such analysis would require a sufficiently large number of data since $O_3$ and CO profiles often depict both a high variability and a complex vertical structure due to the numerous processes at stake. Among the other perspectives, as they combine both chemical (mixing ratios) and dynamical (PBL height) information, these PBL-referenced profiles may offer an interesting way to evaluate the ability of CTMs to properly reproduce the vertical distribution of pollution in a constantly evolving PBL. Although current CTMs are probably not able to reproduce the sharp gradients existing in the capping inversion or entrainment zone due to too coarse vertical resolutions, it remains important to investigate more deeply how well they simulate the vertical distribution of the pollutants under varying PBL conditions. This is particularly important in urban areas where strong emissions occur at a surface characterized by a complex roughness (due to the buildings) which greatly influences the





pollution dispersion. As it operates multi-species ($O_3$, CO, and now $NO_x$, as well as $CO_2$, $CH_4$ and aerosols in the near future) profile measurements in the vicinity of large agglomerations, the IAGOS research infrastructure offers interesting opportunities for such studies. Concerning the representativeness of this dataset and the potential impact of the airport pollution on IAGOS observations, an in-depth $O_3$ and CO comparison between IAGOS and nearby and more

distant surface stations around a few major European airports has recently shown that the IAGOS data in the lowest troposphere depict characteristics typical of urban/suburban surface stations (Petetin et al., 2018). This should encourage more detailed model evaluations of the vertical distribution of the pollution, as now performed operationally with CAMS regional models in the framework of Copernicus (see http://www.iagos.fr/cams for daily comparisons).

**Data availability**

No new measurements were made for this review article. All datasets mentioned in the text were obtained from existing databases. The IAGOS data are available on http://www.iagos.fr or directly via AERIS web site http://www.aeris-data.fr. The ozone soundings can be downloaded from the World Ozone and Ultraviolet Radiation Data Centre (WOUDC) database (http://www.woudc.org) supported by Environment Canada (https://doi.org/10.14287/10000001, WMO/GAW Ozone Monitoring Community, 2018).

**Author contributions**

Contributed to conception and design : HP

Contribution to acquisition of data : HP, VT, BS, HC, GA, RB, DB, J-MC, SR, PN, PN

Contributed to analysis and interpretation of data : HP, BS, HS, FG, FL, VT

Drafted the article : HP

Revised the article : HP, BS, VT, HC, RB, HS, FG

**Competing interests**

The authors have no competing interests to declare.

**Acknowledgements**

We acknowledge the strong support of the European Commission, Airbus, and the Airlines (Lufthansa, Air-France,
Austrian, Air Namibia, Cathay Pacific, Iberia and China Airlines so far) who carry the MOZAIC or IAGOS equipment and perform the maintenance since 1994. In its last 10 years of operation, MOZAIC has been funded by INSU-CNRS (France), Météo-France, Université Paul Sabatier (Toulouse, France) and Research Center Jülich (FZJ, Jülich, Germany). IAGOS has been additionally funded by the EU projects IAGOS-DS and IAGOS-ERI. The MOZAIC-IAGOS database is supported by AERIS (CNES and INSU-CNRS). We also acknowledge the WOUDC, the WMO-
GAW and all individual contributors for providing access to the ozonesonde dataset.

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

**Table 1 : Information relative to the measurements of the parameters used in this study (instrument, uncertainty and period of available data).**

| Type | Parameter | Availability | Measurement technique | Uncertainty |
|---|---|---|---|---|
| IAGOS | RH | 1994-2009 | Capacitive hygrometer | ±5% RH[1] |
|  | $O_3$ | 1994-2016 | Dual-beam UV-absorption monitor | ±2 ppbv / ±2%[2] |
|  | CO | 2002-2016 | Infrared filter correlation instrument | ±5 ppbv / ±5%[3] |
| Ozonesondes | RH | 1994-2016 | Capacitive humidity sensor (usually) | 10% RH[4] |
|  | $O_3$ | 1994-2016 | Electro-chemical Concentration Cell | 3-5%[5] |
|  |  |  | Brewer-Mast | 5-10%[5] |
|  |  |  | Carbon Iodine | 5-10%[5] |

[1] *Helten et al., 1998; Neis et al., 2015a, 2015b.*

[2] *Thouret et al., 1998.*

[3] *Nédélec et al., 2015.*

[4] *Schröder et al., 2017.*

[5] *WMO, 2011.*

**Table 2: Factors of vertical stratification γ (in %) and δ (in % hm$^{-1}$; into brackets) of $O_3$ and CO in the PLB for the different seasons and time slots (see text for details on their calculation).**

| Species | Season | Time of day | | | | | |
|---|---|---|---|---|---|---|---|
|  |  | night | morning | midday | afternoon | daytime | all |
|  | Winter | 10 (2.1) | 11 (1.5) | 11 (1.7) | 12 (1.7) | 11 (1.6) | 11 (1.8) |
| CO | Spring | 9 (1.2) | 10 (1.8) | 8 (1.3) | 10 (1.6) | 9 (1.6) | 9 (1.5) |
|  | Summer | 10 (1.6) | 11 (1.7) | 10 (1.1) | 11 (1.6) | 11 (1.6) | 11 (1.6) |



| | | | | | | | |
|---|---|---|---|---|---|---|---|
| | Fall | 12 (2.3) | 11 (1.7) | 11 (1.4) | 12 (1.4) | 11 (1.5) | 11 (1.8) |
| | Annual | 11 (1.9) | 11 (1.7) | 10 (1.4) | 11 (1.6) | 10 (1.6) | 11 (1.7) |
| | Winter | 25 (7.0) | 21 (5.5) | 18 (3.3) | 21 (3.8) | 20 (4.5) | 21 (5.3) |
| | Spring | 15 (6.1) | 16 (4.8) | 10 (1.3) | 11 (2.0) | 13 (3.4) | 14 (3.9) |
| $O_3$ | Summer | 17 (7.3) | 17 (5.7) | 11 (1.4) | 11 (1.9) | 14 (4.1) | 15 (4.8) |
| | Fall | 26 (9.7) | 22 (6.9) | 17 (2.8) | 18 (2.7) | 20 (4.9) | 21 (6.1) |
| | Annual | 22 (7.6) | 19 (5.7) | 14 (2.2) | 16 (2.6) | 17 (4.2) | 18 (5.1) |



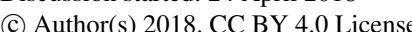

5       **Figure 1: Location of IAGOS airports and WOUDC ozonesonde stations (restricted to 25°N-60°N).**

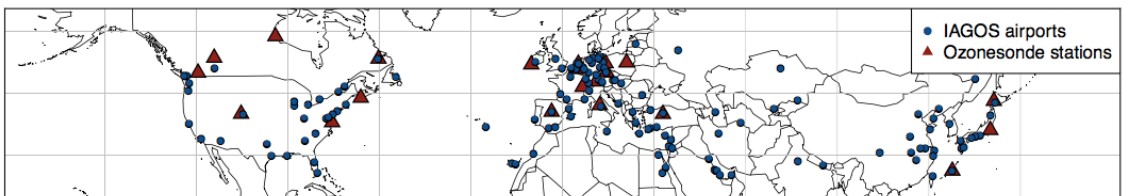

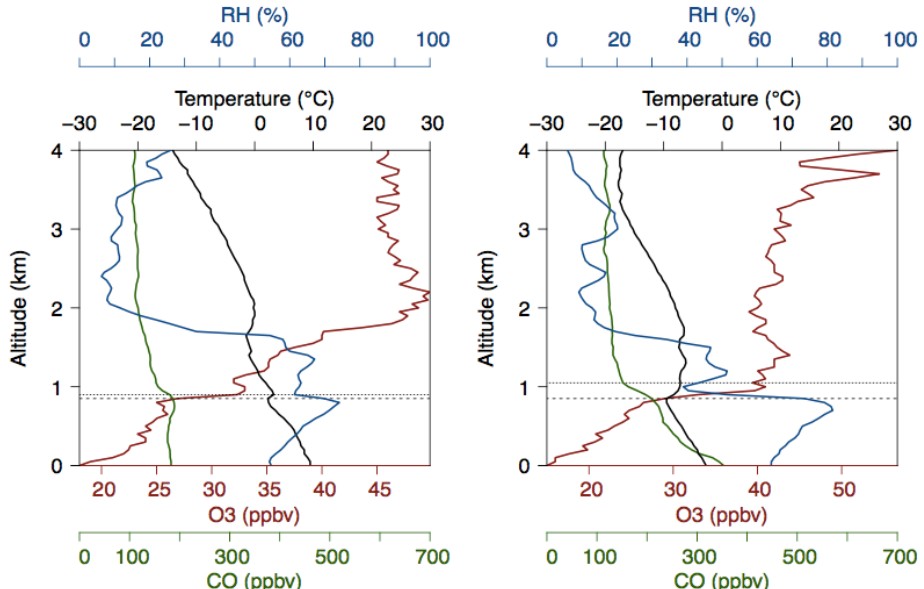

**Figure 2 : Vertical profiles measured by IAGOS on 2004/11/28 (flight n°10275975, left panel) and 2004/12/22 (flight n°10451135, right panel). The curves display the profiles of O₃ (red line) and CO (green line) mixing ratios, RH (blue line) and temperature (black line). The plot also shows the base (dashed black line) and top (dotted black line) of the elevated temperature inversion.**





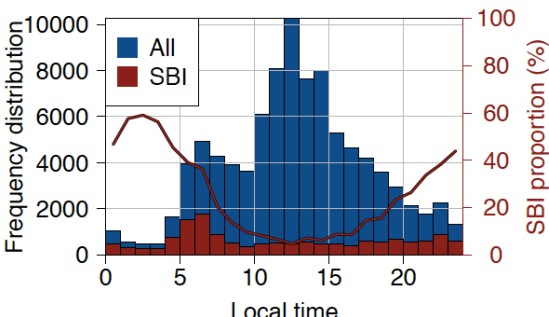

**Figure 3: Frequency distribution of the measurement time of all vertical profiles (blue bars; left axis) and the profiles on which a SBI is detected (red bars superposed over blue ones; left axis). The corresponding proportion of SBI is also plotted (red line; right axis).**

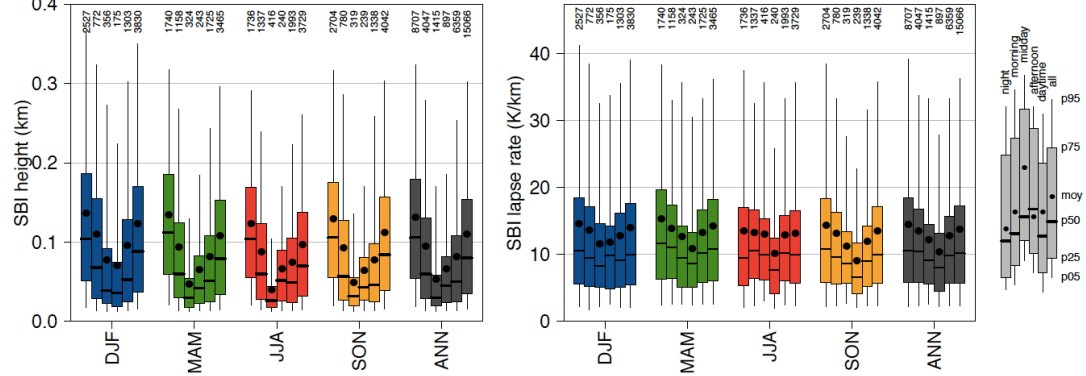

**Figure 4: SBI height (left panel) and temperature lapse rate between the surface and the top of the SBI (right panel). Results are shown for each season (one colour per season) and for the different time slots (adjacent bars from left to right : night, morning, mid-day, afternoon, daytime, entire day; the legend is indicated in grey on the right side). The number of profiles for each bar is indicated on the graph.**



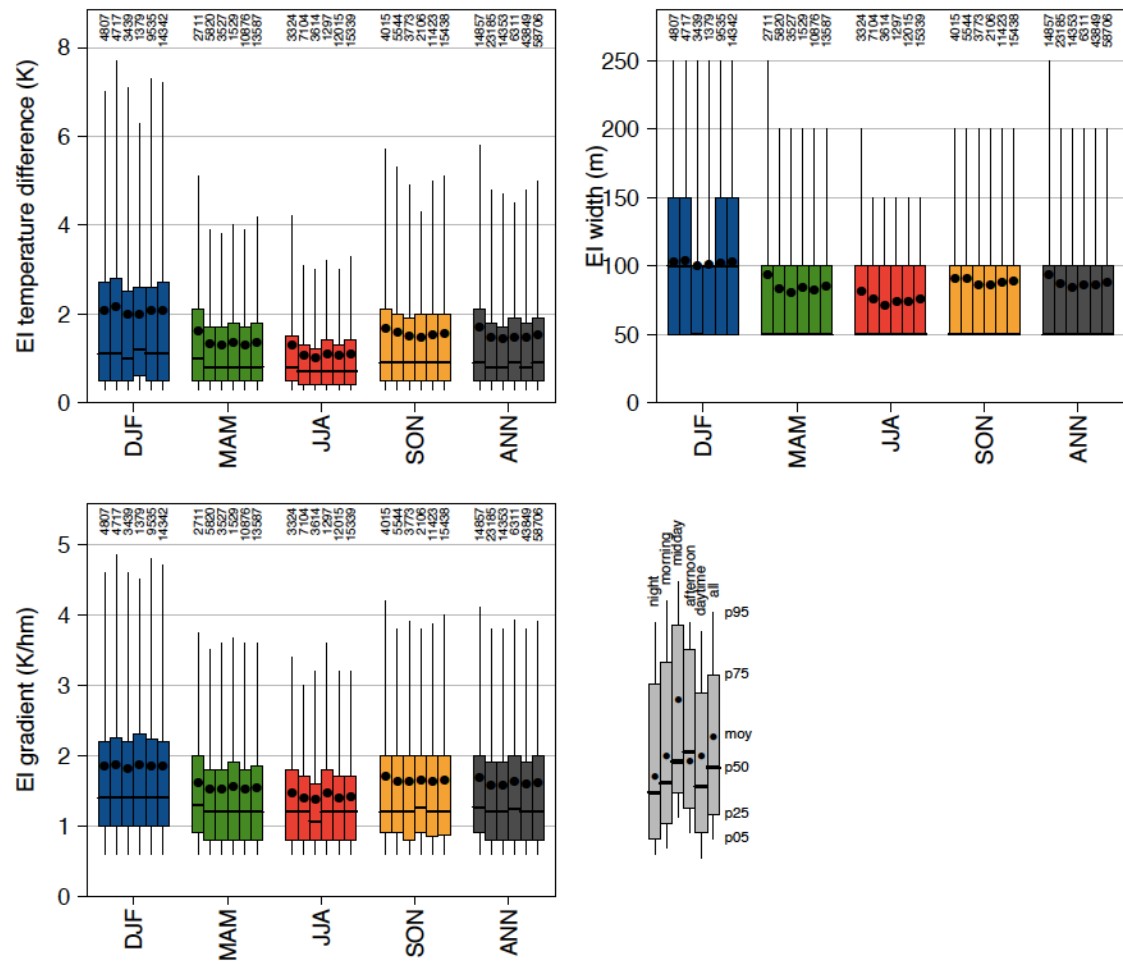

**Figure 5: Diurnal and seasonal variations of the difference of temperature between the EIs' top and base (top left panel), the EI width (top right panel) and the temperature gradient within the EI (bottom left panel). The number of profiles for each bar is indicated on the graph.**




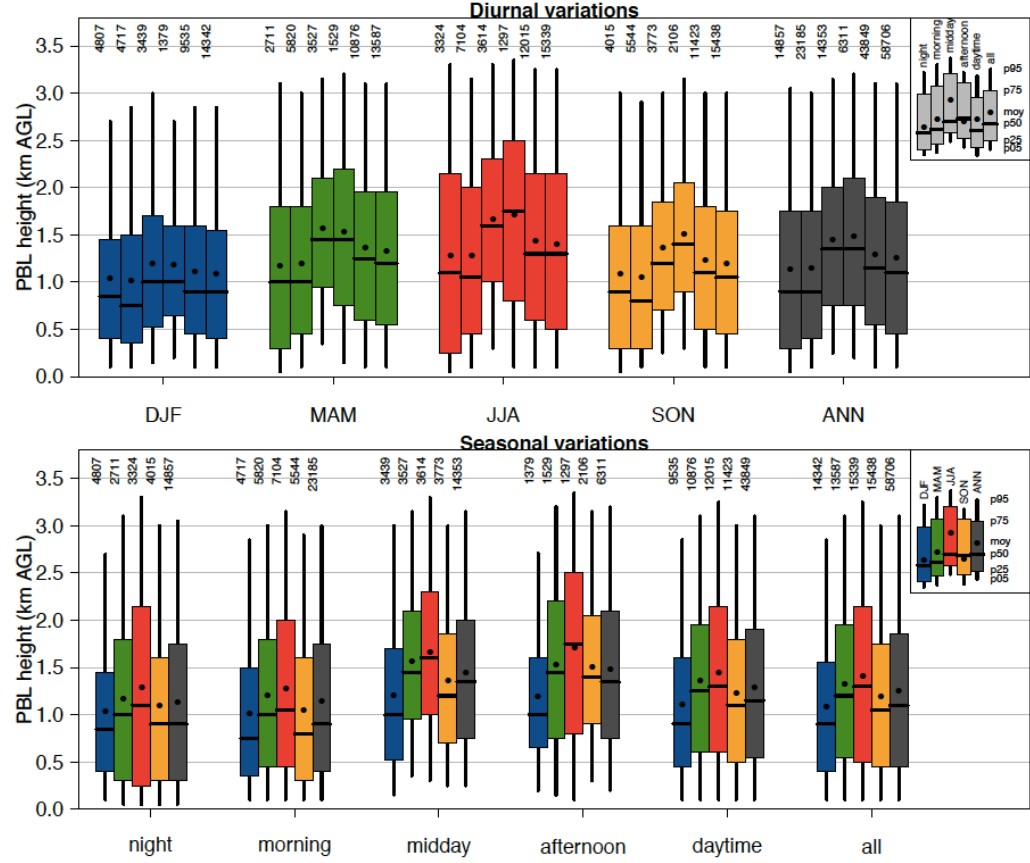

**Figure 6:** Diurnal (top panel) and seasonal (bottom panel) variations of the averaged PBL heights. For all combinations of season and time of the day, the number of available profiles is indicated above the boxplot.



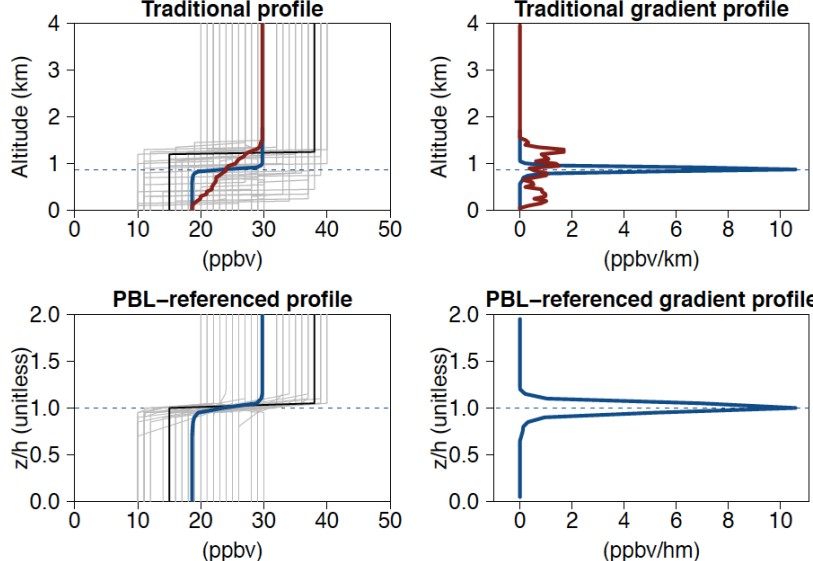

5     **Figure 7: Artificial vertical profiles (left panels) and gradient profiles (right panels) expressed as a function of z (top panels) and z/h (bottom panels). The Figure shows all individual profiles (grey lines) and one example of individual profile (black line), the z-referenced mean profile (red line) and the PBL-referenced profile (blue line) and the mean PBL height of this artificial dataset (blue dotted line) (see text for details).**

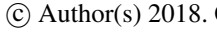



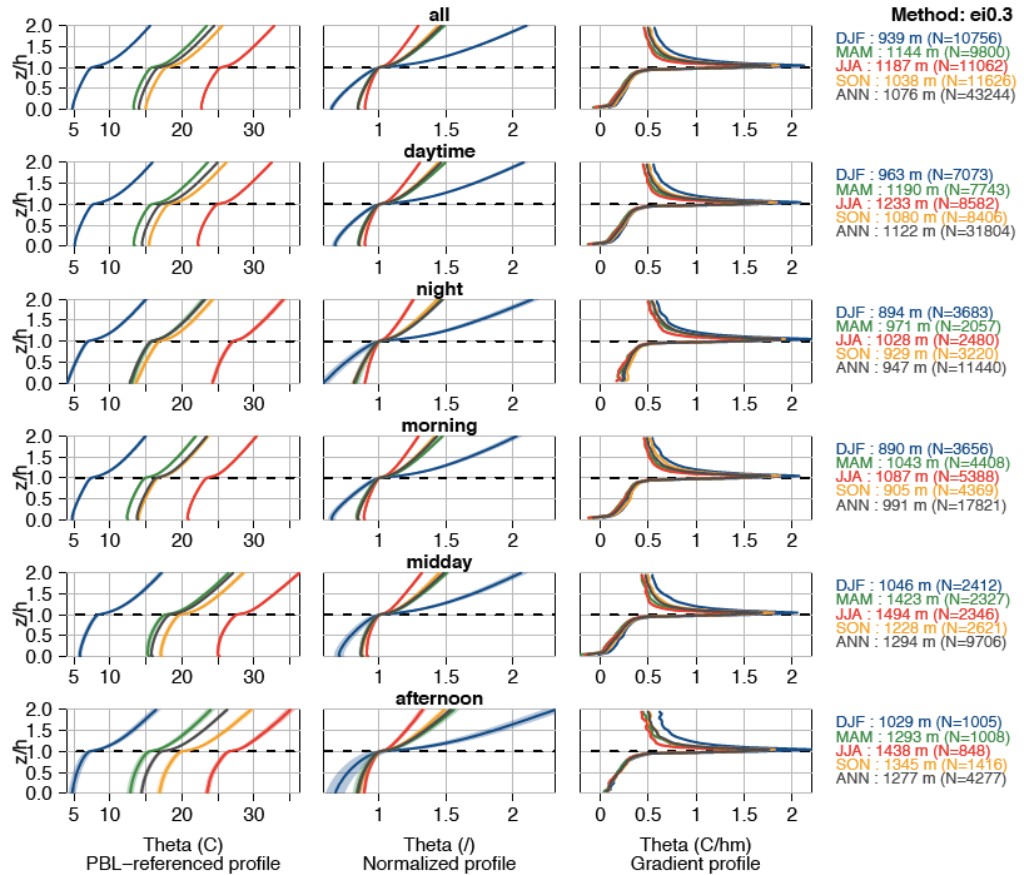

**Figure 8:** Vertical profiles of potential temperature (in °C; left panels), same profiles normalized by the potential temperature at z/h=1 (middle panels), and vertical gradient profiles (in °C hm$^{-1}$; right panels). Plots are shown for different time slots (from top to bottom : all day along, daytime, nighttime, morning, midday, afternoon). The shaded area represents the uncertainties (at a 95% confidence level) on the mean. For each season and time of the day, we indicate the number (N) of profiles used for calculating the PBL-referenced profile (i.e. profiles without any missing data) and the mean PBL height calculated based on this subset of profiles.

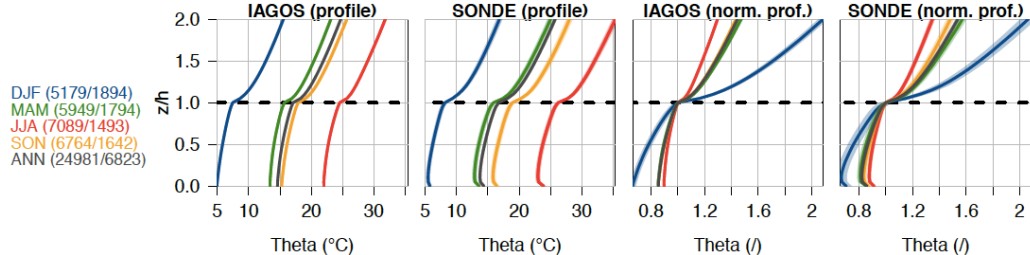

**Figure 9:** IAGOS and ozonesonde PBL-referenced (2 left panels) and normalized profiles (2 right panels) of daytime potential temperature. The number of profiles accounted is indicated for each season (in brackets : IAGOS/SONDE).





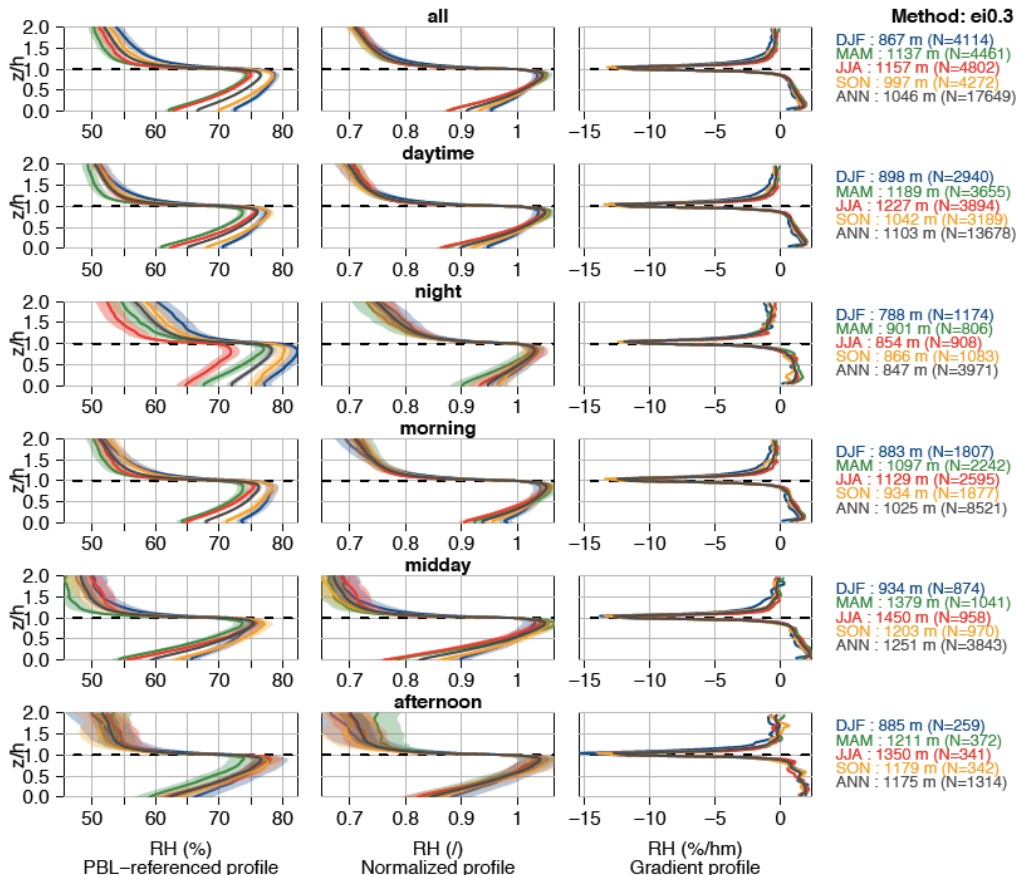

**Figure 10: Same as Fig. 8 for RH (in %).**

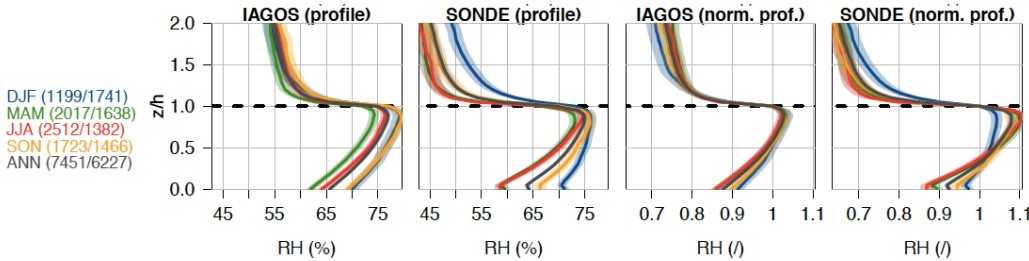

Figure 11: IAGOS and ozonesonde PBL-referenced (2 left panels) and normalized profiles (2 right panels) of daytime RH. The number of profiles accounted is indicated for each season (in brackets : IAGOS/SONDE).





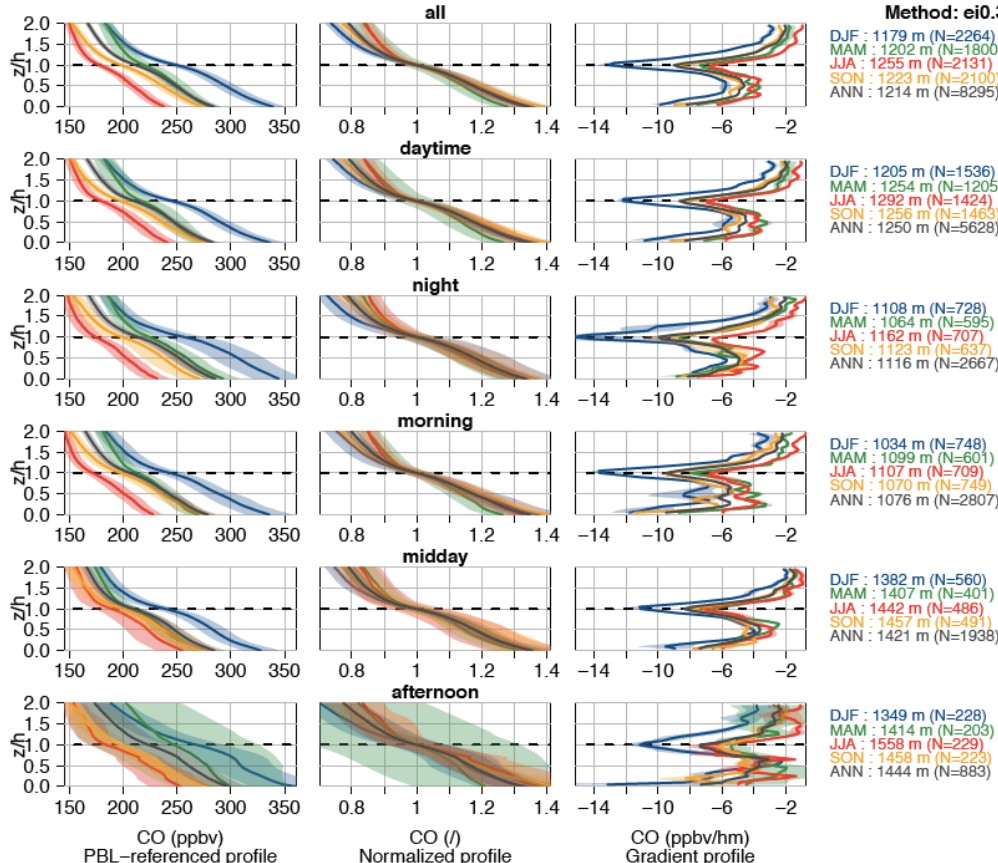

Figure 12: Same as Fig. 8 for CO mixing ratios (in ppbv).





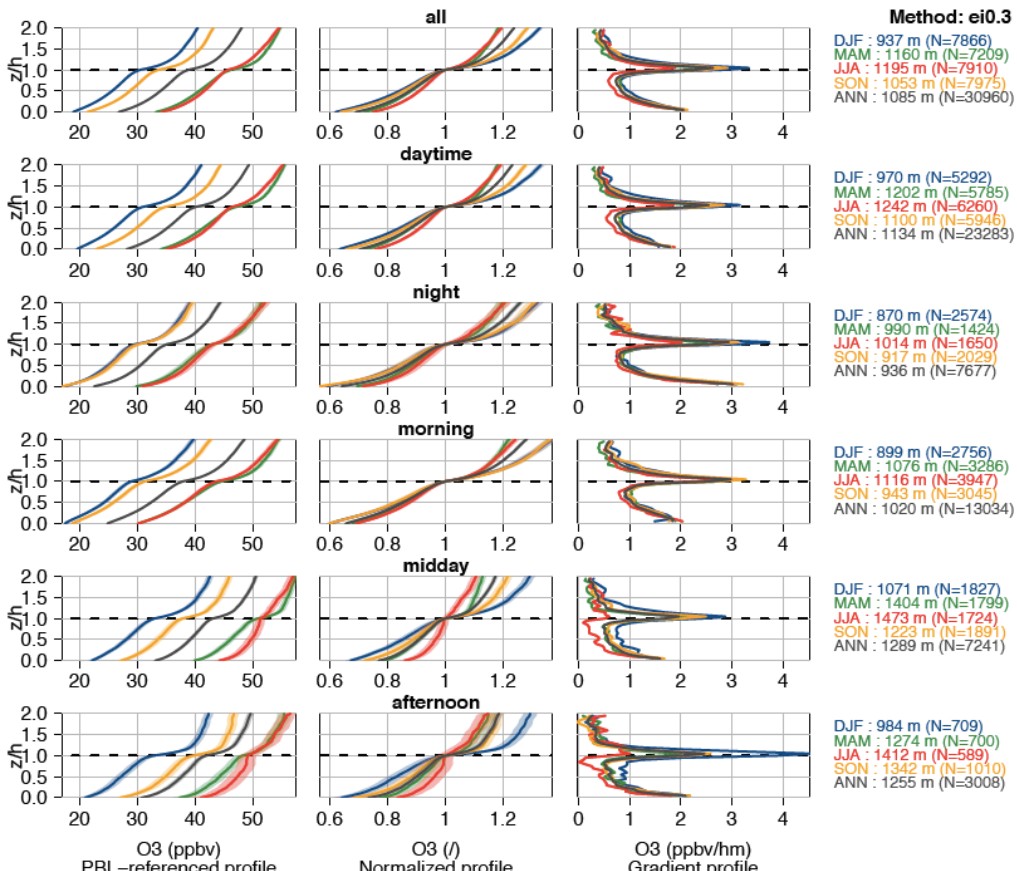

Figure 13: Same as Fig. 8 for O₃ mixing ratios (in ppbv).





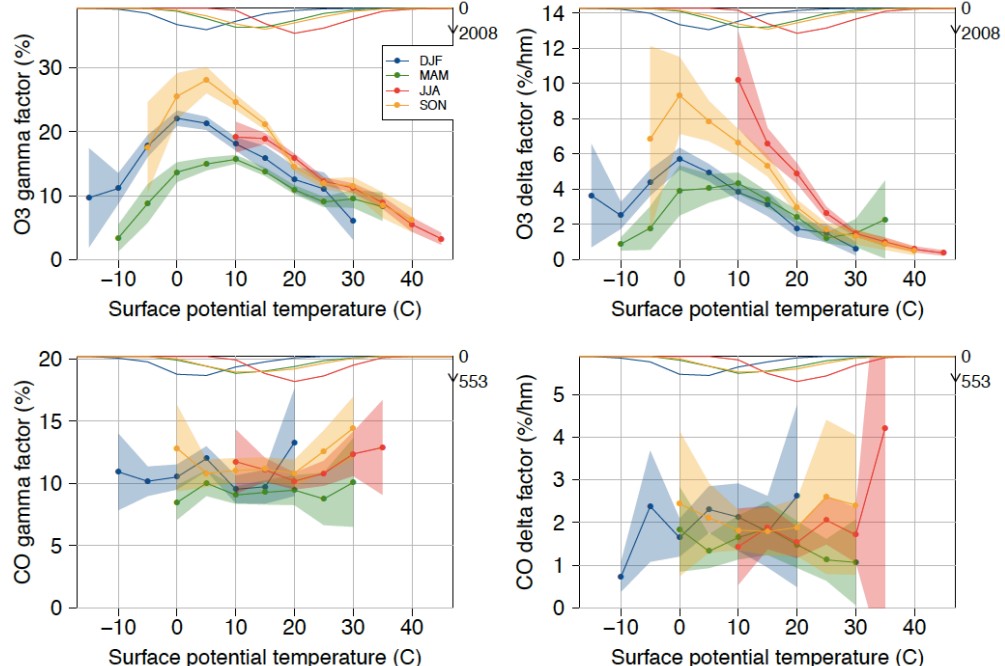

**Figure 14 : Daytime factors of vertical stratification as a function of surface potential temperature. Results are shown for O₃ (top panels) and CO (bottom panels) and for both γ (in %; left panels) and δ (in % hm⁻¹; right panels). The shaded area represents the uncertainties (at a 95% confidence level) on the mean. The curves at the top of the graph show the number of profiles taken into account (in reverse side; the highest number of profile is indicated on the arrow).**

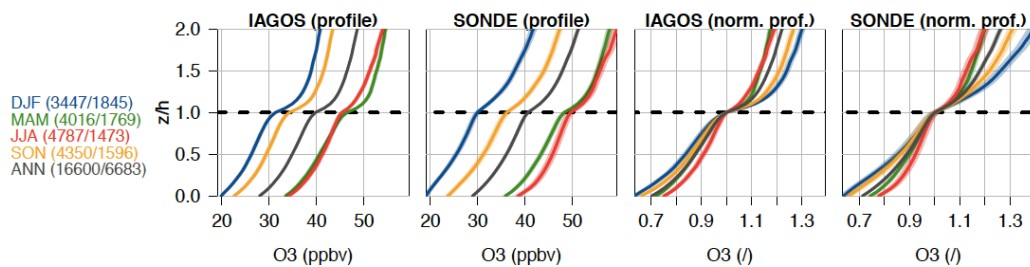

**Figure 15: IAGOS and ozonesonde PBL-referenced (2 left panels) and normalized profiles (2 right panels) of O₃ mixing ratios. The number of profiles accounted is indicated for each season (in brackets : IAGOS/SONDE).**





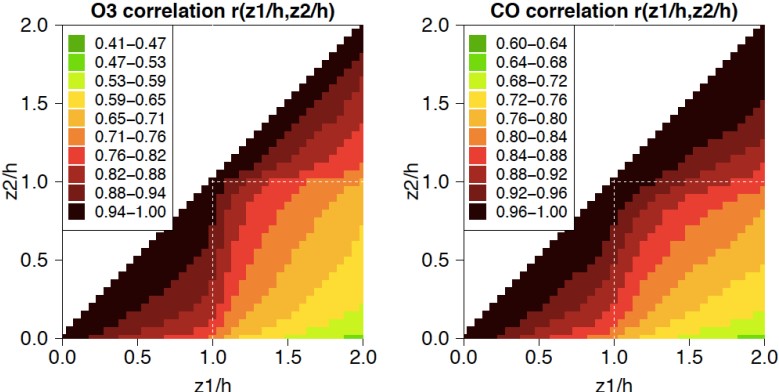

**Figure 16: Correlation of O₃ (left panel) and CO (right panel) mixing ratios between the different z/h altitude levels. All vertical profiles without any data gaps are taken into account. The correlation matrix is symmetric by construction.**