# Peer review of "A climatological view of the vertical stratification of RH, O3 and CO within the PBL and at the interface with free troposphere as seen by IAGOS aircraft and ozonesondes at northern midlatitudes over 1994-2016"

_Atmospheric Chemistry and Physics, 2018_

## Short Comment (SC1) · 24 Apr 2018

Introduction and literature review:

There are studies showing that high-resolution chemistry-climate models with interactive stratospheric and tropospheric chemistry capture the observed layered structure (stratification) of ozone vertical profiles in the free troposphere and at the interface with the PBL. For example, see Figures 3, 5 and 7 in Lin et al. (2012) and Supplementary

[Figure]

Figures 1 and 2 in Lin et al. (2015).

Lin, Meiyun, A. M. Fiore , O. R. Cooper , L. W. Horowitz , A. O. Langford , Hiram Levy II , B. J. Johnson , V. Naik , S. J. Oltmans , C. Senff (2012): Springtime high surface ozone events over the western United States: Quantifying the role of stratospheric intrusions, Journal of Geophysical Research, 117, D00V22, doi:10.1029/2012JD018151

Lin, Meiyun, A.M. Fiore, L.W. Horowitz, A.O. Langford, S. J. Oltmans, D. Tarasick, H.E. Reider (2015): Climate variability modulates western US ozone air quality in spring via deep stratospheric intrusions, Nature Communications, 6, 7105, doi:10.1038/ncomms8105

---

## Referee Comment (RC1) · Anonymous Referee #2 · 18 May 2018

Petetin et al, 2018

The careful development of vertical gradients for ozone and CO with a focus on the boundary layer is a significant contribution to the community. The climatology developed in this work is unique in that the authors took the boundary layer height into account in their averaging. A very interesting finding is that the weakest ozone vertical stratification is observed not only at high temperatures when mixing is expected to be strongest, but also at the lowest observed temperatures, which the authors hypothesize as potentially being due to weak deposition to snow. Finally, this work clearly shows that an aircraft campaign like IAGOS can provide similar profile information to ozonesondes, which is a very useful result that can inform future more extensive CTM evaluation efforts. This manuscript should be published after minor revisions detailed below.

**General Comments**

My main comment is that the climatology developed by the authors should be made publicly available for use in model comparison studies. Currently the paper does not provide a method for obtaining this climatology. In addition, the authors should more carefully discuss the type of PBL they describe in their study due to the use of the EI method. This method would describe a certain type of mixing and exclude other situations and more discussion on the meteorology conditions not captured by this method would be useful. Finally, I would comment that there are a large number of figures and the authors could consider moving figures such as Figure 5 to the supplement that may not actually need more than discussion in the text for the general purposes of the paper.

**Specific Comments**

Page 3, line 14 – If MOZAIC includes NOx, it would be useful to comment on why NOx profiles were not included, also a very useful compound for CTM evaluation particularly due to the sharp gradients in the PBL and the enhancement in the FT due to lightning.

Page 7, line 25 – I think the titles on Figure 6 are swapped – diurnal variation vs seasonal variation seem to be on top of the wrong plots.

Page 8, line 37 – In Figure 8, it looks like theta increases with altitude everywhere. Please clarify. Maybe the resolution on the plot could be increased?

Page 10, line 2 – Could you comment on why the seasonality in RH is not in agreement between IAGOS and the ozonesondes?

Page 10, line 25 – Please explain the % $h m^{-1}$ unit.

Page 12, line 2 – Does the comparison suggest that this climatology is not representative of more polluted regions?

---

## Referee Comment (RC2) · Anonymous Referee #1 · 12 Jun 2018

Major Comments and Recommendation.

Well-done study based on a novel approach to PBL (planetary boundary layer) definitions of EI and SBI. First, variability of these PBLs by season is given. Then, concentrations of ozone, CO and water vapor from IAGOS data are produced within these PBL layers to create climatologies better suited for comparisons to models. In the case of ozone ozonesonde data from stations near IAGOS airports are also used.

The paper should be published after addressing a few comments and after correcting many instances where English grammar or usage needs improvement.

Minor Comments

Page 2, Line 14. It is not accurate to say that in-situ data are lacking at altitude. In addition to IAGOS are literally hundreds of aircraft campaigns over many continents in which ozone, RH, CO profiles have been measured since the late 1980s and early 1990s. The data reside in many open archives such as in European databases and at NASA's Langley and Ames Research Centers.

Page 2, Line 18. GEOS-Chem is considered a CTM (global chemistry-transport model) not an "Air Quality" model. The latter typically has higher horizontal resolution, a more limited vertical range and different approaches to emissions inputs and chemical mechanisms.

Typo/Grammar Fixes on Petetin, Sauvage, Smit et al.

Page 1, Line 14 should have a comma before ozone instead of "and"

Page 1, Line 20 at the end, "novel" is a more appropriate word than original, makes a better impression

Page 1, Line 33 Instead of "Contrary" begin the sentence with "In contrast," ...

Page 1, Line 35 use "in" the presence of snow

Page 2, Line 9 better to day "numerous processes interacting in the PBL"

Page 2, lines 18 and 19. Insert "the" beofre GEOS and before Southeast United States

Page 2, line 23. "The possible error compensations" is awkward and meaning is not clear

Page 2, Line 25. Do you mean "the significance of any conclusions drawn from case studies"?

Page 2, Lines 26-27. Remove "with" on l 26. Line 27 should read "mixing would imply that its..."

Page 2, Line 29 "included" not "including"

Page 2, Line 35. Remove "the" before relative humidity

Page 2, Line 39. Replace "on which" with "from which"

Page 3, Line 6. PBL-FT interface. (End sentence - remove phrase in ())

Page 3, Line 8. "Vertical distributions of O3, CO and RH" - is more clear. At the end of Line 8 modify to "the study and additional perspectives"

Page 3, Line 18. "IAGOS aircraft used in this study are the barometic"

Page 3, Line 30. Calibrated for RH with respect to liquid

Page 3, last line. Uses ozonesonde observations (remove "the")

Page 4, line 4. ("fewer" than 10% not "less" than 10%)

Page 4, Line 7. Factor "of" 3-5

Page 4, Line 24. Less problematic because the vertical variability (not "as")

Page 4, Line 34 spelling United

Page 4, Line 35. Insert "the" Middle East

Page 4, Line 38. "allows smoothing of the vertical" is better

Page 6, Line 8. Allows "us" to maximize the number of profiles taken into account (remove "then")

Page 6, Line 19 Eis are found "in" 16%...

Page 7, Line 15 top "of" the Eis Page 7, Line 20. Characteristics "exhibits" or "displays" is more acceptable word than "depicts"

Page 13. In "Summary and conclusion" standard usage is NOT to refer to Figure numbers again.

Page 13, Line 19 replace "performed" by "archived"

Page 13, Line 21 78% IAGOS profiles, 22% sonde profiles. (Remove "of" and singular sonde, not sondes)

Page 13, Line 23. "Strongly vary throughout the day" not "along"

Page 13, Line 24. "The results" or "Our results"

Page 13, Line 27. .."approach allows us to"

Page 13, Line 30. Eis "displayed" not "depicted"

Page 13, Line 39. Top, "which supports out ability" is correct

Page 14, Line 30. Processes "at work" not "at stake"

Page 14, Line 31. "Interesting way" is weak .... "Better way" or "superior way" or "more meaningful way"

Page 14, Line 34. "resolution" not "resolutions" Replace "deeply" with "thoroughly"

Page 14, Line 39. Replace "interesting" with "rich" or "significant"

---

## Author Comment (AC1) · 20 Jun 2018

**Answers to the first reviewer**

*Major Comments and recommendation*
*Well-done study based on a novel approach to PBL (planetary boundary layer) definitions of EI and SBI. First, variability of these PBLs by season is given. Then, concentrations of ozone, CO and water vapour from IAGOS data are produced within these PBL layers to create climatologies better suited for comparisons to models. In the case of ozone ozonesonde data from stations near IAGOS airports are also used. The paper should be published after addressing a few comments and after correcting many instances where English grammar or usage needs improvement.*

We thank the reviewer for his/her comments and corrections. In the revised manuscript, we took into account all his/her suggestions. In the following, the comments are in blue and the answers in black.

*Minor Comments*
*Page 2, Line 14. It is not accurate to say that in-situ data are lacking at altitude. In addition to IAGOS are literally hundreds of aircraft campaigns over many continents in which ozone, RH, CO profiles have been measured since the late 1980s and early 1990s. The data reside in many open archives such as in European databases and at NASA's Langley and Ames Research Centers.*

Indeed, many aircraft campaigns have been organized over the last decades. However, complete vertical profiles starting from the surface in the lower troposphere are more limited, and in comparison to the surface, the amount of in-situ data in altitude is much lower in altitude. We still modified the sentence as follows : "Over the last decades, a continuous effort was put to collect in-situ observations in the troposphere, mainly with commercial/research aircraft and sondes, and to a lesser extent with instrumented mats and tethered balloons. However, the amount of in-situ data available in altitude remains relatively low compared to the surface (both in terms of quantity of data and number of species). In particular, profiles throughout the entire PBL (i.e. starting from the surface and extending to the free troposphere) are relatively sparse. This limits our ability to properly describe and understand how pollution is vertically distributed within the PBL. One consequence is the difficulty of many state-of-the-art models to reproduce accurately the vertical stratification of the pollution in this part of the troposphere. Although some high-resolution chemistry-climate models (CCMs) with interactive stratospheric and tropospheric chemistry can show encouraging results at the episodic scale (e.g., Lin et al., 2012, 2015), several initiatives of models inter-comparison depicted substantial errors on the ozone ($O_3$) and carbon monoxide (CO) vertical distribution over longer periods of time (Elguindi et al., 2010; Solazzo et al., 2013). More recently, Travis et al. (2017) highlighted the difficulty of the GEOS-Chem chemistry-transport models (CTM) to reproduce sharp $O_3$ vertical gradients in the first kilometre above surface of the Southeast United-States (during both clear-sky and low-cloud conditions), attributed to excessive top-down mixing in the model."

*Page 2, Line 18. GEOS-Chem is considered a CTM (global chemistry-transport model) not an "Air Quality" model. The latter typically has higher horizontal*

*resolution, a more limited vertical range and different approaches to emissions inputs and chemical mechanisms.*
Indeed, we modified the sentence accordingly.

*Typo/Grammar Fixes on Petetin, Sauvage, Smit et al.*
*Page 1, Line 14 should have a comma before ozone instead of "and"*
Modifications applied.

*Page 1, Line 20 at the end, "novel" is a more appropriate word than original, makes a better impression ; Page 1, Line 33 Instead of "Contrary" begin the sentence with "In contrast," ...*
Modifications applied here and in the entire document.

*Page 1, Line 35 use "in" the presence of snow ; Page 2, Line 9 better to say "numerous processes interacting in the PBL" ; Page 2, lines 18 and 19. Insert "the" before GEOS and before Southeast United States*
Modifications applied.

*Page 2, line 23. "The possible error compensations" is awkward and meaning is not clear*
We modified the sentence as follows : "However, a common difficulty in the evaluation of CTMs relies in the fact that several error sources may compensate each other and therefore hide specific model deficiencies. Such error compensations are often complex to identify. In particular, although closely linked, both PBL heights and pollutant concentrations (at the surface and/or along vertical profiles in the PBL) are often evaluated separately, which limits the significance of the drawn conclusions. "

*Page 2, Line 25. Do you mean "the significance of any conclusions drawn from case studies"?*
No we mean that pollutant mixing ratios at the surface and PBL heights are often evaluated separately, usually because no or sparse observations of the PBL height are available for comparisons. This is the case in model evaluations conducted both on the long-term or during case-studies. We simply argue here that as the surface mixing ratios are closely linked to the PBL height (among other parameters), it is tricky to get firm conclusions about the ability of a model to reproduce the surface mixing ratios when its ability to correctly simulate the PBL height has not been assessed simultaneously, or only at one or few locations (that may in addition not correspond to the locations where the chemical composition measurements are performed).

*Page 2, Lines 26-27. Remove "with" on l 26. Line 27 should read "mixing would imply that its..."*
We modified the sentence as follows : "For instance, a model may well reproduce the concentrations of a specific chemical compound at the surface but overestimate the PBL height and/or the vertical mixing ; in this case, this would suggest that its sources are actually overestimated."

*Page 2, Line 29 "included" not "including"*

Modifications applied here and in the entire document.

*Page 2, Line 35. Remove "the" before relative humidity*
Modifications applied.

*Page 2, Line 39. Replace "on which" with "from which"*
We divided the sentence in two parts : "We first implement an algorithm for automatic estimation of the PBL height from both sonde and airborne profiles. Based on these estimates of PBL height, we derive a climatological description of the vertical stratification of the $O_3$, CO, RH and $\theta$ within the PBL and at the interface with the FT. "

*Page 3, Line 6. PBL-FT interface. (End sentence - remove phrase in ()) Page 3, Line 8. "Vertical distributions of O3, CO and RH" - is more clear. At the end of Line 8 modify to "the study and additional perspectives"*
Modifications applied.

*Page 3, Line 18. "IAGOS aircraft used in this study are the barometric"; Page 3, Line 30. Calibrated for RH with respect to liquid*
We modified the sentence as follows : "In this study, we used the barometric altitude, temperature, pressure, calibrated RH with respect to liquid, $O_3$ and CO volume mixing ratios measured on-board IAGOS aircraft."

*Page 3, last line. Uses ozonesonde observations (remove "the") ; Page 4, line 4. ("fewer" than 10% not "less" than 10%) ; Page 4, Line 7. Factor "of" 3-5 ; Page 4, Line 24. Less problematic because the vertical variability (not "as") Page 4, Line 34 spelling United ; Page 4, Line 35. Insert "the" Middle East ; Page 4, Line 38. "allows smoothing of the vertical" is better ; Page 6, Line 8. Allows "us" to maximize the number of profiles taken into account (remove "then") ; Page 6, Line 19 EIs are found "in" 16%... ; Page 7, Line 15 top "of" the EIs*
Modifications applied.

*Page 7, Line 20. Characteristics "exhibits" or "displays" is more acceptable word than "depicts"*
Modifications applied here and in the entire document.

*Page 13. In "Summary and conclusion" standard usage is NOT to refer to Figure numbers again.*
We removed them.

*Page 13, Line 19 replace "performed" by "archived"*
We replaced "performed" by "measured".

*Page 13, Line 21 78% IAGOS profiles, 22% sonde profiles. (Remove "of" and singular sonde, not sondes) ; Page 13, Line 23. "Strongly vary throughout the day" not "along" ; Page 13, Line 24. "The results" or "Our results" Page 13, Line 27. .."approach allows us to" ; Page 13, Line 30. EIs "displayed" not "depicted" ; Page 13, Line 39. Top, "which supports out ability" is correct ; Page 14, Line 30. Processes "at work" not "at stake" ; Page 14, Line 31. "Interesting way" is weak .... "Better*

*way" or "superior way" or "more meaningful way" ; Page 14, Line 34. "resolution" not "resolutions" Replace "deeply" with "thoroughly" ; Page 14, Line 39. Replace "interesting" with "rich" or "significant"*

Modifications applied.

---

## Author Comment (AC2) · 20 Jun 2018

**Answers to the second reviewer**

*The careful development of vertical gradients for ozone and CO with a focus on the boundary layer is a significant contribution to the community. The climatology developed in this work is unique in that the authors took the boundary layer height into account in their averaging. A very interesting finding is that the weakest ozone vertical stratification is observed not only at high temperatures when mixing is expected to be strongest, but also at the lowest observed temperatures, which the authors hypothesize as potentially being due to weak deposition to snow. Finally, this work clearly shows that an aircraft campaign like IAGOS can provide similar profile information to ozonesondes, which is a very useful result that can inform future more extensive CTM evaluation efforts. This manuscript should be published after minor revisions detailed below.*

We greatly thank the reviewer for his/her positive feedback on our study. In the following, the comments are in blue and the answers in black.

*General Comments*

*My main comment is that the climatology developed by the authors should be made publicly available for use in model comparison studies. Currently the paper does not provide a method for obtaining this climatology.*

We agree that it would be useful for the modelling community to easily and freely access this climatology. For both $O_3$ and CO, the PBL-referenced profiles for all seasons and times of day are now provided as text files and properly identified with a DOI (http://dx.doi.org/10.25326/4). The DOI gives access to a brief description of these products. They are now publicly available on the IAGOS website.

We added page 15 line 3 : "In order to allow further studies, the mean climatological $O_3$ and CO PBL-referenced profiles analysed in this study are freely available on the IAGOS portal (http://dx.doi.org/10.25326/4), for each season and time of day."

In the data availability section : "The climatological $O_3$ and CO PBL-referenced profiles are available through the IAGOS central database (http://iagos.sedoo.fr) and are part of the ancillary products (http://dx.doi.org/10.25326/4) (Petetin et al., 2018b)."

And in the references : "Petetin, H., Sauvage, B. and Boulanger, D.: IAGOS ancillary data: PBL-referenced profiles of $O_3$ and CO, 2018b."

*In addition, the authors should more carefully discuss the type of PBL they describe in their study due to the use of the EI method. This method would describe a certain type of mixing and exclude other situations and more discussion on the meteorology conditions not captured by this method would be useful.*

In this study, we are excluding only the profiles with SBIs (16% of the profiles), the profiles with vertical differences of temperature below 0.3 K (13%) and the profiles that do not meet the quality criteria in terms of data gaps (8%). The PBL characteristics analysed here with the EI method are finally based on 63% of the profiles (58,706 profiles). Therefore, apart from the very specific situations of atmospheric stability at the surface, most of the available profiles are taken into account here.

*Finally, I would comment that there are a large number of figures and the authors could consider moving figures such as Figure 5 to the supplement that may not actually need more than discussion in the text for the general purposes of the paper.*

We agree with the reviewer that there is a quite large number of figures in this paper. As suggested, we thus moved Fig. 5 in the Supplement. We modified the sentence page 7 line 17 as follows : "We investigated some characteristics of the EIs, namely the temperature difference, width and temperature vertical gradient (see Fig. S-1 in the Supplement)."

However, we think that the other figures should be included in the main document. A reader more interested in the vertical stratification of one specific chemical compound or thermodynamical parameter can still easily skip some parts and go directly to the figures and text of interest for him.

*Specific Comments*
*Page 3, line 14 – If MOZAIC includes NOx, it would be useful to comment on why NOx profiles were not included, also a very useful compound for CTM evaluation particularly due to the sharp gradients in the PBL and the enhancement in the FT due to lightning.*

MOZAIC only included NOy measurements on-board one aircraft between 2001 and 2005. Conversely, IAGOS now includes NOx measurements (currently on-board one aircraft) but these data are not available yet due to on-going validation. Even for NOy, the amount of data available in the lower troposphere is considerably lower than for CO or $O_3$ (there are many data gaps in the profile in the first kilometres). In the near future, when a sufficient number of NOx profiles will be available, we fully agree that it will be interesting to investigate its vertical stratification. Note that there are some studies in preparation in the IAGOS group with the new NOx observations.

*Page 7, line 25 – I think the titles on Figure 6 are swapped – diurnal variation vs seasonal variation seem to be on top of the wrong plots.*

No, actually both panels in Figure 6 shows the same results but organized in a different way. For instance, the top panel depicts, for each season, the diurnal variations : the box-and-whisker for the different times of day are gathered in order to highlight the diurnal variations (and this is done for all seasons). This is why we denominate it "Diurnal variations".

*Page 8, line 37 – In Figure 8, it looks like theta increases with altitude everywhere. Please clarify. Maybe the resolution on the plot could be increased?*

As explained in the text, for some seasons and times of day, the potential temperature is decreasing with altitude but very weakly and only between the two first altitudes (i.e. z/h between 0 and 0.05). Although it is possible to see it, we agree that it is difficult to see (because the decrease is very weak). However, we do think there is no easy way to modify the figure in order to highlight this small feature more clearly (since we are trying to keep figures reasonably compact). But this is one of the reasons for which we also showing the vertical

gradient profiles (right panels) where this decrease of the potential temperature with altitude is obvious.

*Page 10, line 2 – Could you comment on why the seasonality in RH is not in agreement between IAGOS and the ozonesondes?*

Looking at the RH PBL-referenced profiles in the PBL, we can see that the vertical gradients of RH differ between IAGOS and ozonesonde (as mentioned in the text), but the seasonality remains in reasonable agreement with lower RH during spring/summer and higher RH during winter/fall. In contrast, larger differences are observed in the lower free troposphere. In this part of the troposphere, in contrast to IAGOS that depicts relatively similar RH during all seasons, ozonesondes display lower RH values. This is particularly true during spring/summer and to a lesser extent during fall but the differences are reduced for winter. This results in a change of seasonality compared to IAGOS. To our opinion, this may be explained by the already mentioned negative bias on RH sonde measurements due to the heating of the sensors by the solar radiation. As solar radiations are strongest during spring/summer and lowest during winter, this would be consistent with the seasonal differences observed here.

We added a sentence : "This is also supported by the fact the differences between IAGOS and sondes are largely reduced when considering only nighttime profiles, i.e. when radiosonde measurements are not affected by heating effects due to solar radiation (not shown). These sources of bias are also expected to vary from one season to the other following the seasonality of solar radiation that are strongest in spring/summer and lowest in winter/fall. This may (at least partly) explain the distortion of the seasonal variations of RH in ozonesondes compared to IAGOS in the lower free troposphere. "

*Page 10, line 25 – Please explain the % hm$^{-1}$ unit.*

This is already explained page 8, lines 33-35. However, we modified page 7, line 22-23 (the first occurrence of "hm") : "This leads to mean temperature vertical gradients of 1.4 and 1.9 K hm$^{-1}$ during these two seasons, respectively. Interestingly, none of these characteristics depicts a diurnal variation (whatever the statistical metric)." → "This leads to mean temperature vertical gradients of 1.4 and 1.9 K hm$^{-1}$ (where hm stands for hectometre, i.e. 100 m) during these two seasons, respectively. Interestingly, none of these characteristics depicts a diurnal variation (whatever the statistical metric)."

*Page 12, line 2 – Does the comparison suggest that this climatology is not representative of more polluted regions?*

Yes the reviewer is right. More precisely, it may not be representative of the most polluted regions during episodes of $O_3$ pollution. We added this sentence in the text : "[…] thus in good agreement with Kaser et al. (2017). This suggests that our climatology may not be representative of the most polluted regions during $O_3$ pollution episodes. […] "

---

## Author Comment (AC3) · 20 Jun 2018

**Answers to Meiyun Lin**

*Introduction and literature review:*

*There are studies showing that high-resolution chemistry-climate models with interactive stratospheric and tropospheric chemistry capture the observed layered structure (stratification) of ozone vertical profiles in the free troposphere and at the interface with the PBL. For example, see Figures 3, 5 and 7 in Lin et al. (2012) and Supplementary.*

*Figures 1 and 2 in Lin et al. (2015). Lin, Meiyun, A. M. Fiore , O. R. Cooper , L. W. Horowitz , A. O. Langford , Hiram Levy II , B. J. Johnson , V. Naik , S. J. Oltmans , C. Senff (2012): Springtime high surface ozone events over the western United States: Quantifying the role of stratospheric intrusions, Journal of Geophysical Research, 117, D00V22, doi:10.1029/2012JD018151*

*Lin, Meiyun, A.M. Fiore, L.W. Horowitz, A.O. Langford, S. J. Oltmans, D. Tarasick, H.E. Reider (2015): Climate variability modulates western US ozone air quality in spring via deep stratospheric intrusions, Nature Communications, 6, 7105, doi:10.1038/ncomms8105.*

We thank Meiyun Lin for these additional references. Indeed these two studies show that the GFDL AM3 model captures reasonably well the vertical structure of $O_3$ profiles. However, to our opinion, the time and space coverage of these specific studies remain too limited to get firm conclusions on the model ability to reproduce the $O_3$ vertical gradients in the lower troposphere. But we agree that a good representation of the chemistry combined with a high spatial resolution is very useful to get closer to the observed profiles of $O_3$ mixing ratios. We included these two references in the manuscript, and modified the paragraph page 2 lines 15-21 as followed : "Over the last decades, a continuous effort was put to collect in-situ observations in the troposphere, mainly with commercial/research aircraft and sondes, and to a lesser extent with instrumented mats and tethered balloons. However, the amount of in-situ data available in altitude remains relatively low compared to the surface (both in terms of quantity of data and number of species). In particular, profiles throughout the entire PBL (i.e. starting from the surface and extending to the free troposphere) are relatively sparse. This limits our ability to properly describe and understand how pollution is vertically distributed within the PBL. One consequence is the difficulty of many state-of-the-art models to reproduce accurately the vertical stratification of the pollution in this part of the troposphere. Although some high-resolution chemistry-climate models (CCMs) with interactive stratospheric and tropospheric chemistry can show encouraging results at the episodic scale (e.g., Lin et al., 2012, 2015), several initiatives of models inter-comparison depicted substantial errors on the ozone ($O_3$) and carbon monoxide (CO) vertical distribution over longer periods of time (Elguindi et al., 2010; Solazzo et al., 2013). More recently, Travis et al. (2017) highlighted the difficulty of the GEOS-Chem chemistry-transport models (CTM) to reproduce sharp $O_3$ vertical gradients in the first kilometre above surface of the Southeast United-States (during both clear-sky and low-cloud conditions), attributed to excessive top-down mixing in the model."